# DynaMath: A Dynamic Visual Benchmark for Evaluating Mathematical Reasoning Robustness of Vision Language Models

**Chengke Zou**[1,2*†]**, Xingang Guo**[1*]**, Rui Yang**[1*]**, Junyu Zhang**[1]**, Bin Hu**[1]**, Huan Zhang**[1]

[1]University of Illinois at Urbana-Champaign, [2]University of California, Berkeley
`chengke_zou@berkeley.edu`,`{xingang2,ry21,junyuz6}@illinois.edu`
`binhu7@illinois.edu, huan@huan-zhang.com`
**Project page:** https://dynamath.github.io

## Abstract

The rapid advancements in Vision-Language Models (VLMs) have shown great potential in tackling mathematical reasoning tasks that involve visual context. Unlike humans who can reliably apply solution steps to similar problems with minor modifications, we found that state-of-the-art VLMs like GPT-4o can consistently fail in these scenarios, revealing limitations in their mathematical reasoning capabilities. In this paper, we investigate the **mathematical reasoning robustness** in VLMs and evaluate how well these models perform under different variants of the same question, such as changes in visual numerical values or function graphs. While several vision-based math benchmarks have been developed to assess VLMs' problem-solving capabilities, these benchmarks contain only static sets of problems and cannot easily evaluate mathematical reasoning robustness. To fill this gap, we introduce DynaMath, a dynamic visual math benchmark designed for in-depth assessment of VLMs. DynaMath includes 501 high-quality, multi-topic *seed* questions, *each represented as a Python program*. Those programs are carefully designed and annotated to enable the automatic generation of a much larger set of *concrete* questions, including many different types of visual and textual variations. DynaMath allows us to evaluate the generalization ability of VLMs, by assessing their performance under varying input conditions of a seed question. We evaluated 14 state-of-the-art VLMs with 5,010 generated concrete questions (10 per seed question). Our results show that the worst-case model accuracy, defined as the percentage of correctly answered seed questions in all 10 variants, is significantly lower than the average-case accuracy. In addition, many models show high consistency in answering these questions – the incorrectness of a certain variant of a seed question is not only due to inherent randomness. Our analysis emphasizes the need to study the robustness of VLMs' reasoning abilities, and DynaMath provides valuable insights to guide the development of more reliable models for mathematical reasoning.

## 1 Introduction

Leveraging pretraining on vast Internet-scale datasets, Large Language Models (LLMs) (Brown, 2020; Ouyang et al., 2022; Touvron et al., 2023; Achiam et al., 2023) and Multi-modal Large Language Models (MLLMs) (Team et al., 2023; Bai et al., 2023; Liu et al., 2024c;a) have achieved remarkable performance across a wide range of tasks. Among them, Vision-Language Models (VLMs) (Zhu et al., 2023; Zhang et al., 2024b) stand out, showing exceptional promise as versatile assistants capable of integrating vision and language for problem-solving.

Among their visual comprehension abilities across different domains, mathematical reasoning (Lightman et al., 2023; Zhang et al., 2024f) stands out as a crucial measure of human-like intelligence, requiring both math knowledge and logical thinking. Recent work has proposed many benchmarks for evaluating the mathematical reasoning ability of VLMs. MATHVISTA (Lu et al., 2023)

---

[*]Equal contribution.
[†]Work done during internship at UIUC.

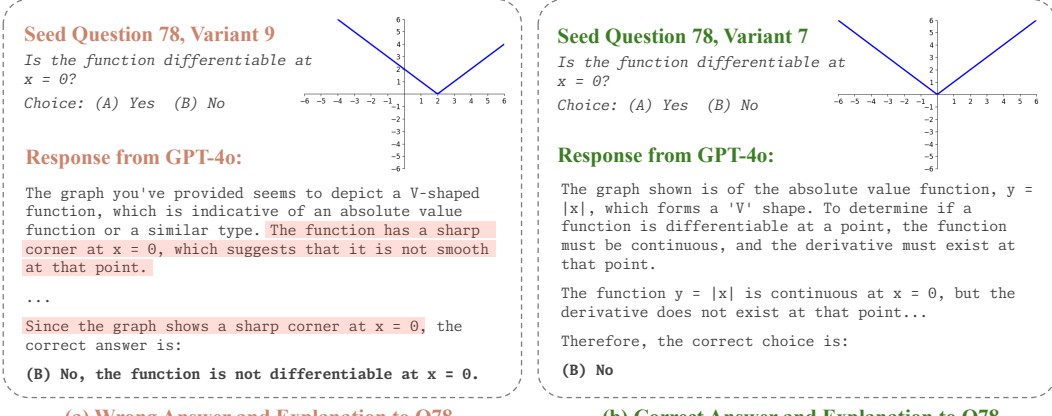

(a) Wrong Answer and Explanation to Q78       (b) Correct Answer and Explanation to Q78

Figure 1: An example of consistent failures in GPT-4o. Seed question 78 in our DYNAMATH benchmark generates a graph of a shifted absolute value function. GPT-4o consistently provides incorrect answers for variant 9 **(left)** with 90% repetition consistency, while it can successfully answer variant 7 **(right)** with 100% repetition consistency. We tested for other 8 variants involving non-zero shifts of the absolute value function, GPT-4o insists that the "sharp corner" is at $x = 0$ and produces an incorrect answer for 7 variants. More failure examples are in Appendix F.

was the first benchmark specifically designed to evaluate visual mathematical reasoning. Recent closed-source models, such as Claude 3.5 Sonnet and GPT-4o, along with open-source models like LLaVA-OneVision (Li et al., 2024), have demonstrated average performance surpassing that of humans. Benchmarks such as MATH-V (Wang et al., 2024a) and MATHVERSE (Zhang et al., 2024e) demonstrate the current limitations of VLMs in handling challenging mathematical problems and understanding mathematical diagrams. Following typical evaluation pipelines, these benchmarks contain a *static* set of testing questions on which a VLM will be scored.

Our work is inspired by recent studies (Nezhurina et al., 2024; Zheng et al., 2023; Zong et al., 2023; Mirzadeh et al., 2024), which found that even powerful LLMs struggle to reliably solve simple text reasoning problems under different input values or conditions. We found that this issue is even more pronounced in VLMs due to the added complexity of visual context. In the setting of math problems, we identified consistent failure cases on *variations of simple questions*. As illustrated in Figure 1, we identify a simple question asking whether a shifted absolute value function $f(x) = |x - a|$ is differentiable at $x = 0$. Despite the shift, this question is still quite simple and poses no challenges to humans. While GPT-4o can give correct answers for some values of $a$, it consistently gives a wrong answer for many different values of $a \neq 0$. Drawing inspiration from human reasoning, where the same steps can be applied to solve similar problems with varying conditions, a robust reasoning model should exhibit the same ability. This raises important questions about the robustness of VLMs' reasoning abilities: ***are the reasoning procedures in VLMs robust to problem variations that pose no challenge to humans?***

To address this question, we comprehensively study the robustness of mathematical reasoning in VLMs by introducing a new benchmark, DYNAMATH. DYNAMATH is a *dynamic* visual math benchmark designed for an in-depth assessment of VLMs' reasoning robustness. Unlike existing benchmarks, which contain a static dataset of benchmarking questions, DYNAMATH contains 501 high-quality *seed* questions covering multiple mathematical topics: Plane Geometry, Solid Geometry, Analytic Geometry, Algebra, Puzzle Tests, Graph Theory, Statistics, Scientific Figures, and Arithmetic. Each seed question is represented as a carefully designed Python *program*; upon running, a program generates diverse *concrete* instances of one seed question with random variations in its conditions. The program is individually written for each seed question and considers multiple possible types of variations in each question, such as variations of numerical values, function types, graph structure, geometry, mathematical operations, etc. The questions also span varying difficulty levels, from elementary school to high school and undergraduate, with the latter two dominating. The process of dynamic benchmark generation and evaluation is presented in Figure 2. During evaluation, many concrete questions are created from a single seed question, and thus the actual number of questions evaluated can be much greater (e.g., $10\times$ more) than the number of seed questions.

We conducted extensive experiments on DYNAMATH to evaluate the reasoning robustness of current state-of-the-art (SOTA) closed-source models, including GPT-4o, Gemini Pro, and Claude-3.5

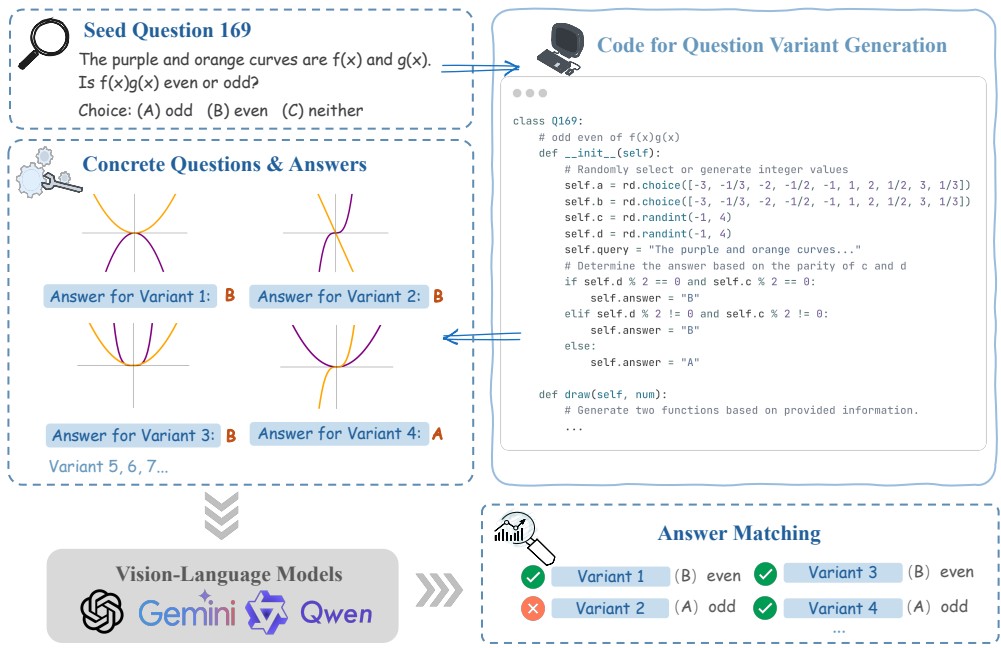

Figure 2: The dynamic benchmark generation procedure in DYNAMATH. A seed question is represented as a program that can generate many concrete questions with different variations. The plots for concrete questions are randomly generated along with the corresponding ground-truth answers. During evaluation, all concrete variants of the seed questions are considered, allowing us to evaluate the worst-case model performance and robustness.

Sonnet, as well as open-source VLMs such as the InternVL2 series (Chen et al., 2024), LLaVA-v1.6 series (Liu et al., 2024b), Qwen2-VL (Wang et al., 2024c), DeepSeek-VL (Lu et al., 2024), and Llama 3.2 (Dubey et al., 2024). For each seed problem, we randomly generated 10 variants, resulting in an evaluation dataset of 5,010 concrete problems. On these problems, we evaluate both average-case accuracy and worst-case accuracy. The *worst-case accuracy* is defined as the percentage of correctly answered seed problems in *all* 10 variants. We observe that all considered VLMs have a worst-case accuracy that is close to or less than 50% of the average-case accuracy, signifying their unreliability in handling question variations. In addition, we also evaluate the *repetition consistency* on these VLMs, which characterizes the model randomness to ensure that a low worst-case accuracy is not solely caused by occasional random errors but also consistent errors on certain variants of a seed problem. Our main contributions and findings can be summarized as:
• We are the first to study the mathematical reasoning robustness of VLMs and identified a new weakness in VLMs: they may consistently fail on certain variants of simple math questions that pose no challenges to humans. Such a weakness is prevalent in many state-of-the-art VLMs.
• We introduce DYNAMATH, a dynamic benchmark comprising 501 individually designed programs capable of generating a large number of question variants across different types. Our work is the first dynamically generated benchmark for evaluating the math capability of VLMs.
• Based on 5,010 concrete questions generated by DYNAMATH, we conduct an extensive evaluation of both SOTA closed-source and open-source VLMs. We find a noticeable gap between the average-case accuracy and worst-case accuracy among all models, indicating that many VLMs do not have robust reasoning capabilities even on relatively simple mathematical questions.

## 2 RELATED WORK

**Mathematical Reasoning Benchmarks.** Reasoning ability is a key indicator of intelligence, prompting researchers to develop various benchmark datasets to assess the mathematical reasoning capabilities of LLMs and VLMs. Numerous benchmarks have been proposed for evaluating this ability in the text-only domain, including (Amini et al., 2019; Hendrycks et al., 2020; 2021; Cobbe et al., 2021; Mishra et al., 2022; Frieder et al., 2024; Yu et al., 2023; Zhang et al., 2024a). Additionally, recent research has begun to shift its focus towards the evaluation of robustness and the creation of dynamic benchmarks for language models. Several studies (Stolfo et al., 2022; Wu et al., 2023; Srivastava et al., 2024; Nezhurina et al., 2024; Qian et al., 2024; Kurtic et al., 2024; Mirzadeh et al., 2024) assess the language models' robustness to the changing of item names or value conditions in the text-based question. However, many real-world problems, such as those involving

statistical charts and geometry, rely on visual context. To assess visual mathematical reasoning, several benchmarks have been designed around geometry tasks (Lu et al., 2021; Chen et al., 2021) or multiple-choice questions (Liu et al., 2023; Yue et al., 2024). Among these, Liu et al. (2023) studied the robustness of VLMs when faced with changes in the order of multiple-choice questions. Recent efforts have expanded these benchmarks to cover a broader array of topics and question formats, such as MATHVISTA (Lu et al., 2023), MATHVERSE (Zhang et al., 2024e), and MATH-V (Wang et al., 2024a). Despite the diverse range of questions and visual contexts in these benchmarks, they share a common limitation: both the visual components and text remain static. This allows models to potentially achieve high scores by memorizing patterns from the training data, rather than applying true reasoning skills. In contrast, this paper introduces DYNAMATH, a dynamic visual math benchmark that provides a more rigorous assessment of VLMs' reasoning capabilities through dynamically generating math questions with visual content.

**Vision-Language Models (VLMs)**   With the success of LLMs, numerous closed-source VLMs, such as GPT-4o, Gemini, and Claude 3.5, have excelled across a variety of visual-based understanding and conversational tasks, highlighting the potential of multimodal AI assistants. In the open-source domain, several efforts are actively advancing the field. Approaches like LLaMA-Adapter (Zhang et al., 2024d; Gao et al., 2023) and MiniGPT-4 (Zhu et al., 2023) leverage frozen language models with a limited number of trainable parameters, demonstrating promising results. Furthermore, a range of VLMs trained on larger multimodal datasets has been open-sourced, pushing the frontier of visual comprehension and generalization ability. Notable examples include the InternVL1.5 and InternVL2 series (Chen et al., 2024), InternLM-XComposer (Zhang et al., 2023; Dong et al., 2024), LLaVA-v1.6 series (Liu et al., 2024b), LLaVA-OneVision (Li et al., 2024), Qwen-VL (Bai et al., 2023; Wang et al., 2024c), and DeepSeek-VL (Lu et al., 2024). These models contribute significantly to advancing the capabilities of VLMs in prior visual benchmarks.

## 3 BENCHMARK DESIGN

We present DYNAMATH, a curated evaluation dataset aimed at assessing the robustness of visual language models (VLMs) in multimodal mathematical reasoning across a wide variety of mathematical tasks with dynamic visual and textual contexts.

### 3.1 DATASET COLLECTION

Our benchmark collection comprises two phases: seed question collection and program-based question generation. In the initial phase, we selectively curate a set of high-quality mathematics problems that necessitate reasoning based on visual information. The subsequent phase involves transforming each seed question into code-based prototypes, allowing for the generation of diverse concrete questions under randomly sampled conditions.

**Seed question Collection.**   The seed questions are sourced from existing visual math datasets and publicly available online resources. We identify 107 questions from MathVista (Lu et al., 2023), covering fundamental concepts in analytic geometry, planar geometry, and statistics. Additionally, we source 27 questions from MATH-V (Wang et al., 2024a), which serve as prototypes for topics related to arithmetic, puzzle tests, and solid geometry. To augment the dataset's breadth and depth, we included 45 questions based on scientific figures and 48 undergraduate-level questions focused on graph theory, drawn from the MMMU dataset (Yue et al., 2024) and various accessible educational materials. Furthermore, we incorporated 236 questions requiring advanced reasoning on topics such as functions, geometry, and statistics, all gathered from publicly available resources on the internet. To diversify the question types represented in our collection, we also developed 38 new problems by ourselves covering linear algebra, set theory, and algorithmic flow.

Following the collection of seed questions, we conducted a comprehensive review to eliminate any questions that included excessively complex images, as these would pose challenges for programmatic generation. Ultimately, as shown in Figure 4(b), our benchmark consists of 501 seed questions, with 227 (45.3%) sourced from established visual math datasets, while 274 (54.7%) are newly collected or developed from public resources.

Note that our goal is not to create the most challenging, competition-level benchmark as in (Wang et al., 2024a), but rather to provide relatively easy benchmarks with diverse variants to evaluate robustness. Nonetheless, we ensure that the difficulty of our questions is comparable to the levels of datasets such as MATHVERSE (Zhang et al., 2024e) and MATHVISTA (Lu et al., 2023).

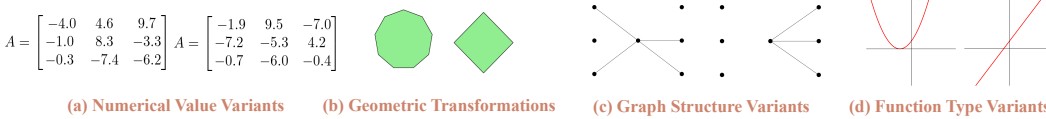

Figure 3: Examples of variation types in DYNAMATH. More examples are in Appendix B and D.

**Program-based Question Generation.** After establishing our seed questions, we recruited a group of college STEM students to annotate each question with the common strategies they employed in solving them. These annotations served as prototypes for developing corresponding programs tailored to each question. As illustrated in Figure 2, each question is represented as a carefully crafted Python program, which encompasses a defined range of conditions for sampling and algorithmic calculations to derive the solution. Additionally, we implemented a drawing function in each program, utilizing libraries such as Matplotlib and Pyglet to generate corresponding images based on varying conditions. Specifically, 470 of the question programs incorporate a plotting function that leverages the randomly sampled conditions to create the visual context of the question, while the remaining 31 question programs utilize fixed images, randomizing only the textual elements. This programmatic approach allows the generation of a large number of concrete benchmark questions by executing the generation program multiple times, facilitating the efficient creation of new problems and enabling the evaluation of the reasoning robustness of VLMs.

As shown in Figure 3, we integrate various types of variants to enrich the diversity of question generation for DYNAMATH:

1. **Numerical Value Variants**: Modifying numerical quantities to evaluate the VLM's proficiency in handling different numerical values and performing arithmetic operations.

2. **Geometric Transformations**: Altering shapes, angles, dimensions, and relative positions to examine the spatial and geometric understanding of VLMs.

3. **Function Type Variants**: Varying different types of mathematical functions (e.g., linear, quadratic) to evaluate how well models generalize across functional representations.

4. **Color Variants**: Changing object or curve colors randomly to test the model's recognition of visual patterns and its robustness to superficial alterations.

5. **Symbolic Substitutions**: Modifying symbolic elements such as mathematical operations to determine the model's adaptability to various symbolic representations.

6. **Graph Structure Variants**: Modifying graph layouts, networks, or other structural representations to assess the model's comprehension of relationships and topological features.

7. **Real-life Contexts Variants**: Adjusting the contents of real-world scenarios (e.g., calendars, time-related problems, or poker-like questions) to test the model's contextual understanding and application to practical situations.

Each variant category targets a specific facet of mathematical reasoning, making DYNAMATH a comprehensive benchmark for evaluating the flexibility, robustness, and accuracy of VLMs in solving mathematical problems. Detailed diagrams of each variation are provided in Appendix B.

### 3.2 DATASET STATISTICS

Detailed statistics on the data composition of DYNAMATH are presented in Table 1. DYNAMATH encompasses nine mathematical topics: Solid Geometry (SG, 3.0%), Puzzle Tests (PT, 3.4%), Arithmetic (AR, 5.2%), Scientific Figures (SF, 9.0%), Graph Theory (GT, 9.6%), Algebra (AL, 10.2%), Plane Geometry (PG, 15.4%), Analytic Geometry (AG, 19.4%), and Statistics (ST, 25.0%). Examples for each topic are provided in Appendix D. Each topic necessitates a nuanced understanding of image context, foundational mathematical knowledge, practical reasoning abilities, and logical deduction skills. Importantly, the dataset is designed to cater to varying levels of difficulty, ranging from elementary to undergraduate education, with a notable focus on high school (55.3%) and undergraduate (32.1%) levels. In terms of question types, the dataset consists of 59.1% numerical questions, 34.7% multiple-choice questions, and 6.2% free-form questions. While VLMs might occasionally answer multiple-choice questions correctly by chance, free-form questions provide a more precise evaluation of the model's capabilities. Consequently, our dataset emphasizes free-form questions, distinguishing it from previous visual math benchmarks such as MATHVISTA (Lu et al., 2023), MATHVERSE (Zhang et al., 2024e), and MATH-V (Wang et al., 2024a), which predominantly include more than 50% multiple-choice questions.

| Statistic | Number |
|---|---|
| Total *seed* questions (programs) | 501 |
| - Created from existing dataset | 227 (45.3%) |
| - Newly designed questions | 274 (54.7%) |
| Topics | |
| - Solid geometry (SG) | 15 (3.0%) |
| - Puzzle test (PT) | 17 (3.4%) |
| - Arithmetic (AR) | 26 (5.2%) |
| - Scientific figure (SF) | 45 (9.0%) |
| - Graph theory (GT) | 48 (9.6%) |
| - Algebra (AL) | 51 (10.2%) |
| - Plane geometry (PG) | 77 (15.4%) |
| - Analytic geometry (AG) | 97 (19.4%) |
| - Statistics (ST) | 125 (25.0%) |
| Levels | |
| - Elementary school (EL) | 63 (12.6%) |
| - High school (HI) | 277 (55.3%) |
| - Undergraduate (UN) | 161 (32.1%) |
| Question Types | |
| - Numerical questions | 296 (59.1%) |
| - Multiple-choice questions | 174 (34.7%) |
| - Free-form questions | 31 (6.2%) |

Table 1: Statistics of DYNAMATH.

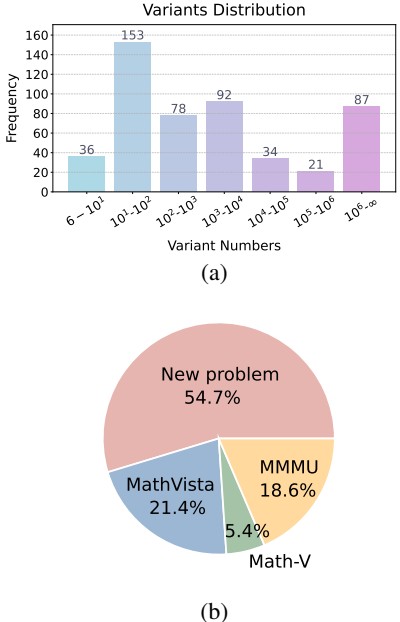

Figure 4: (a) Variant number distribution and (b) source composition of DYNAMATH.

In Figure 4(a), we illustrate the distribution of variant numbers among the 501 seed questions. Notably, approximately 30.5% of the seed questions have a possible variant number ranging from 10 to $10^2$. Nearly 93% of the seed questions contain more than 10 variants, and 17.4% of them have more than $10^6$ potential variants, demonstrating the diversity of variations in our dataset.

### 3.3 EVALUATION PROTOCOLS

Our evaluation process consists of two stages: answer extraction and score calculation. Following the methodology of prior work (Lu et al., 2022), we utilize prompt engineering and template matching to extract answers. Prompts guide the model to generate responses in both full and short answer formats. After generation, the short answer is extracted for comparison with the ground truth. Detailed prompts used in our experiments can be found in Appendix C.

Our dataset contains $N = 501$ seed questions in total. For each seed question in the dataset, we generate $M = 10$ variants, resulting in a total of $5,010$ concrete questions. We evaluate two metrics: **average-case accuracy** ($\mathcal{A}_{avg}$) and **worst-case accuracy** ($\mathcal{A}_{wst}$) over these variants. The two metrics are different from prior benchmarks that evaluate only a single instance of a question. The metrics are defined as follows:

$$\mathcal{A}_{avg} = \frac{1}{N} \sum_{i=1}^{N} \frac{1}{M} \sum_{j=1}^{M} \mathbb{I}[\text{Ans}(i,j) = \text{GT}(i,j)], \ \mathcal{A}_{wst} = \frac{1}{N} \sum_{i=1}^{N} \min_{j \in [1,M]} \mathbb{I}[\text{Ans}(i,j) = \text{GT}(i,j)],$$
(1)

where $\text{Ans}(i,j)$ and $\text{GT}(i,j)$ represent the generated answer and the ground truth answer for variant $j$ of question $i$. We also define **Reasoning Robustness** ($RR$) as the ratio between the average-case performance and the worst-case performance:

$$RR = \frac{\mathcal{A}_{wst}}{\mathcal{A}_{avg}},$$
(2)

The model's response uncertainty reflects both the impact of input changes and inherent uncertainty, the latter of which can be represented by the concept of repetition consistency ($RC$), similar to self-consistency (Wang et al., 2022). We define repetition consistency as:

$$RC(i,j) = \frac{1}{K} \sum_{k=1}^{K} \mathbb{I}[\text{Ans}_k(i,j) = \text{Ans}(i,j)],$$
(3)

where $K$ is number of repetitions and $\text{Ans}_k(i,j)$ is the $k$-th repetition for $j$-th variant of $i$-th seed question. The repetition consistency represents the model's confidence in the answer $\text{Ans}(i,j)$.

Table 2: Average-case accuracy $\mathcal{A}_{avg}$ on DYNAMATH with 5,010 generated questions. "ALL" represents overall accuracy. Question topics and difficulty levels (PG, EL, etc) are defined in Table 1.

| Model | ALL | PG | SG | AG | AL | PT | GT | ST | SF | AR | EL | HI | UN |
|---|---|---|---|---|---|---|---|---|---|---|---|---|---|
| *Closed-sourced Large Multimodal Models (LMMs)* | | | | | | | | | | | | | |
| Zero-shot GPT-4o | 63.7 | 56.8 | 52.0 | 61.0 | 76.9 | 51.8 | 58.1 | 69.3 | 62.4 | 61.5 | 68.6 | 61.8 | 36.8 |
| Zero-shot Claude-3.5 | 64.8 | 49.9 | 49.3 | 55.3 | 81.0 | 44.1 | 69.4 | 78.2 | 62.2 | 61.2 | 66.7 | 62.6 | 33.3 |
| Zero-shot Gemini Pro 1.5 | 60.5 | 52.7 | 42.7 | 61.6 | 70.8 | 20.6 | 65.2 | 69.8 | 50.2 | 54.2 | 62.9 | 59.2 | 37.1 |
| 3-shot CoT GPT-4o | 64.9 | 58.1 | 59.3 | 57.7 | 84.1 | 51.2 | 61.9 | 71.0 | 60.9 | 57.7 | 66.2 | 62.5 | 34.8 |
| 3-shot CoT Claude-3.5 | 62.5 | 49.1 | 48.0 | 50.6 | 80.2 | 37.1 | 58.1 | 78.2 | 64.9 | 55.0 | 63.0 | 61.5 | 30.5 |
| 3-shot CoT Gemini Pro 1.5 | 58.7 | 52.6 | 45.3 | 56.7 | 72.9 | 21.8 | 57.9 | 66.0 | 54.9 | 48.1 | 59.0 | 58.3 | 34.2 |
| *Open-source Vision Language Models (VLMs)* | | | | | | | | | | | | | |
| Qwen2-VL-72B | 55.1 | 48.1 | 48.7 | 50.9 | 57.6 | 28.2 | 45.0 | 68.9 | 56.4 | 54.2 | 61.3 | 57.4 | 30.7 |
| Qwen2-VL-72B (3-shot CoT) | 52.4 | 45.1 | 44.7 | 47.5 | 59.4 | 19.4 | 44.2 | 67.1 | 52.9 | 53.1 | 61.0 | 53.6 | 28.6 |
| Qwen2-VL-7B | 42.1 | 40.3 | 38.7 | 39.9 | 37.1 | 8.2 | 44.8 | 52.1 | 41.1 | 39.2 | 47.6 | 42.2 | 24.4 |
| InternVL2-76B | 54.0 | 44.5 | 34.7 | 43.8 | 67.6 | 35.3 | 51.0 | 66.7 | 55.1 | 51.5 | 60.3 | 52.9 | 26.4 |
| InternVL2-40B | 41.8 | 31.3 | 21.3 | 38.8 | 42.9 | 15.3 | 38.3 | 58.1 | 43.1 | 38.1 | 51.0 | 41.5 | 23.4 |
| InternVL2-26B | 41.0 | 35.8 | 26.0 | 37.3 | 38.8 | 13.5 | 46.9 | 51.9 | 39.6 | 40.4 | 52.1 | 38.5 | 22.5 |
| InternVL2-8B | 39.7 | 33.9 | 37.3 | 32.5 | 46.9 | 15.9 | 42.1 | 47.8 | 39.1 | 37.3 | 51.1 | 37.4 | 19.6 |
| Llama-3.2-90B | 44.0 | 47.5 | 37.3 | 36.8 | 46.5 | 12.4 | 44.8 | 56.8 | 39.8 | 30.0 | 45.4 | 43.8 | 22.2 |
| Deepseek-VL-7B-chat | 21.5 | 16.0 | 13.3 | 26.5 | 12.9 | 4.7 | 32.7 | 24.3 | 24.2 | 15.0 | 28.3 | 19.0 | 16.0 |
| Llava-v1.6-34B | 27.1 | 21.4 | 25.3 | 27.6 | 14.9 | 7.6 | 32.7 | 36.8 | 27.8 | 23.1 | 35.9 | 23.8 | 16.6 |
| Llava-v1.6-vicuna-13B | 19.8 | 14.7 | 10.0 | 23.4 | 8.2 | 10.0 | 21.5 | 28.2 | 19.6 | 10.0 | 27.1 | 16.5 | 14.1 |
| Llava-v1.5-7B | 16.6 | 10.5 | 7.3 | 19.5 | 6.5 | 8.2 | 32.3 | 17.5 | 20.2 | 10.8 | 18.9 | 13.3 | 11.7 |
| *Human* | | | | | | | | | | | | | |
| Human performance | 77.3 | 79.9 | 66.7 | 80.4 | 77.5 | 73.5 | 69.8 | 78.0 | 78.9 | 75.0 | 78.6 | 79.8 | 72.7 |

## 4 EXPERIMENT

In this section, we conduct thorough experiments to assess the performance and reasoning robustness of various closed-source and open-source models on the DYNAMATH dataset. Subsequently, we present detailed quantitative results and qualitative analyses in Sections 4.2 and 4.3, respectively.

### 4.1 EXPERIMENTAL SETUPS

We evaluate the performance of two sets of models on the DYNAMATH benchmark, which involves 10 variations for each seed question, resulting in a total of 5010 questions. The first group comprised SOTA closed-source VLMs, such as GPT-4o, Gemini Pro 1.5, and Claude-3.5 Sonnet, with zero-shot and 3-shots with Chain-of-Thought (CoT) configurations. The second group consisted of SOTA open-source VLMs, including Qwen2-VL (7B, 72B) (Wang et al., 2024c), InternVL2 (8B, 26B, 40B, 76B) (Chen et al., 2024), Llava-v1.5 (7B) (Liu et al., 2024a), Llava-v1.6 (13B, 34B) (Liu et al., 2024b), Deepseek-VL (7B) (Lu et al., 2024), and Llama 3.2 (90B) (Dubey et al., 2024). We specifically explored open-source models with varying parameter sizes to analyze the impact of model size on reasoning robustness. The deployment of open-source models relied on the `lmdeploy` package (Contributors, 2023). We set the temperature to 0.0 for all models to reduce inherent randomness. Details regarding the prompts and hyperparameters used in this experiment are outlined in Appendix C.

To assess human performance, we generated a new variant dataset consisting of 1002 concrete questions (2 variants per seed question). These questions were divided into 20 questionnaires, each containing 50 to 51 questions. We then recruited 20 undergraduates or graduates from STEM to help establish the baseline for human performance based on their average scores.

For the few-shot setup, we follow the standard approach by including three demonstration examples, each accompanied by the associated visual elements. Given the diverse range of topics covered in DYNAMATH, we provide topic-specific demonstration examples to ensure its relevance for each problem in DYNAMATH. Specifically, we curated five demonstration examples from MathVista (Lu et al., 2023) and MathVision (Wang et al., 2024a) for each topic, and then randomly select three examples when evaluating DYNAMATH problems within the corresponding topic. In addition, we incorporate detailed reasoning steps in the demonstration examples, following a typical Chain-of-Thought (CoT) setup (Wei et al., 2022). Detailed demonstrations and prompts in Appendix C.3.

### 4.2 EXPERIMENTAL RESULTS

In this section, we present a detailed comparison of the top-performing VLMs on DYNAMATH, as shown in Table 2 and Table 3.

**Overall Results on Average Accuracy.** Table 2 illustrates the average-case performance of a variety of closed-source and open-source models. Within the closed-source category, GPT-4o, Claude-

Table 3: Worst-case accuracy $\mathcal{A}_{wst}$ on DYNAMATH with 5,010 generated questions. "ALL" represents overall accuracy. Question topics and difficulty levels (PG, EL, etc) are defined in Table 1.

| Model | ALL | PG | SG | AG | AL | PT | GT | ST | SF | AR | EL | HI | UN |
|---|---|---|---|---|---|---|---|---|---|---|---|---|---|
| *Closed-sourced Large Multimodal Models (LMMs)* | | | | | | | | | | | | | |
| Zero-shot GPT-4o | 34.7 | 37.7 | 33.3 | 25.8 | 54.9 | 11.8 | 18.8 | 38.4 | 35.6 | 46.2 | 46.0 | 34.3 | 31.1 |
| Zero-shot Claude-3.5 | 35.3 | 22.1 | 26.7 | 18.6 | 62.7 | 23.5 | 27.1 | 53.6 | 24.4 | 42.3 | 49.2 | 33.2 | 33.5 |
| Zero-shot Gemini Pro 1.5 | 26.9 | 28.6 | 20.0 | 19.6 | 39.2 | 5.9 | 22.9 | 35.2 | 15.6 | 30.8 | 41.3 | 26.7 | 21.7 |
| 3-shot CoT GPT-4o | 32.3 | 31.2 | 40.0 | 21.6 | 54.9 | 17.6 | 20.8 | 36.8 | 26.7 | 46.2 | 47.6 | 30.7 | 29.2 |
| 3-shot CoT Claude-3.5 | 32.1 | 27.3 | 26.7 | 11.3 | 54.9 | 0.0 | 10.4 | 56.0 | 31.1 | 30.8 | 39.7 | 32.9 | 28.0 |
| 3-shot CoT Gemini Pro 1.5 | 23.6 | 27.3 | 26.7 | 14.4 | 39.2 | 5.9 | 18.8 | 27.2 | 17.8 | 26.9 | 33.3 | 23.1 | 20.5 |
| *Open-sourced Vision Language Models (VLMs)* | | | | | | | | | | | | | |
| Qwen2-VL-72B | 28.3 | 27.3 | 33.3 | 15.5 | 31.4 | 0.0 | 16.7 | 43.2 | 26.7 | 42.3 | 41.3 | 30.3 | 19.9 |
| Qwen2-VL-72B (3-shot COT) | 22.8 | 24.7 | 26.7 | 8.2 | 35.3 | 0.0 | 8.3 | 32.8 | 22.2 | 38.5 | 41.3 | 23.5 | 14.3 |
| Qwen2-VL-7B | 13.8 | 22.1 | 6.7 | 7.2 | 13.7 | 0.0 | 12.5 | 16.8 | 11.1 | 19.2 | 25.4 | 12.3 | 11.8 |
| InternVL2-76B | 24.6 | 24.7 | 20.0 | 15.5 | 37.3 | 5.9 | 12.5 | 32.8 | 20.0 | 38.5 | 39.7 | 23.1 | 21.1 |
| InternVL2-40B | 14.2 | 14.3 | 6.7 | 9.3 | 13.7 | 0.0 | 10.4 | 21.6 | 13.3 | 19.2 | 28.6 | 14.1 | 8.7 |
| InternVL2-26B | 14.4 | 19.5 | 0.0 | 6.2 | 9.8 | 0.0 | 18.8 | 20.0 | 11.1 | 26.9 | 34.9 | 12.3 | 9.9 |
| InternVL2-8B | 10.4 | 13.0 | 20.0 | 5.2 | 15.7 | 0.0 | 10.4 | 9.6 | 11.1 | 15.4 | 23.8 | 9.4 | 6.8 |
| Llama-3.2-90B | 13.0 | 22.1 | 20.0 | 7.2 | 7.8 | 0.0 | 12.5 | 16.8 | 13.3 | 3.8 | 15.9 | 14.1 | 9.9 |
| Deepseek-VL-7B-chat | 4.2 | 7.8 | 0.0 | 3.1 | 0.0 | 0.0 | 10.4 | 4.0 | 2.2 | 3.8 | 7.9 | 2.9 | 5.0 |
| Llava-v1.6-34B | 6.0 | 10.4 | 13.3 | 4.1 | 2.0 | 0.0 | 4.2 | 6.4 | 6.7 | 7.7 | 15.9 | 5.1 | 3.7 |
| Llava-v1.6-vicuna-13B | 2.8 | 7.8 | 0.0 | 4.1 | 0.0 | 0.0 | 2.1 | 2.4 | 0.0 | 0.0 | 6.3 | 2.9 | 1.2 |
| Llava-v1.5-7B | 1.8 | 3.9 | 0.0 | 2.1 | 0.0 | 0.0 | 4.2 | 0.8 | 0.0 | 3.8 | 3.2 | 1.8 | 1.2 |

3.5, and Gemini Pro 1.5 exhibit average accuracies higher than 60%, with Claude-3.5 achieving the highest zero-shot average accuracy at 64.8%. However, there remains an 12.5% disparity when compared to human performance, which stands at 77.3%. This highlights the need for further development in the reasoning ability of VLMs. Regarding the 3-shot CoT performance, it is intriguing to note that there is no consistent improvement across different closed-sourced models, confirming findings from previous research (Wang et al., 2024a). For instance, while 3-shot CoT GPT-4o manages to enhance zero-shot performance from 63.7% to 64.9%, both 3-shot CoT Claude-3.5 and 3-shot CoT Gemini Pro 1.5 experience a decline in performance (64.8% → 62.5% and 60.5% → 58.7% respectively). Moving on to the open-sourced models, although they generally underperform when compared to closed-sourced models, the gap has been narrowed by recent models such as Qwen2 and InternVL2, which have more than 70B parameters. This noteworthy progress is evident when comparing them to previous benchmark results like MATHVISTA (Amini et al., 2019), MATHVERSE (Zhang et al., 2024e), and MATH-V (Wang et al., 2024a). It highlights the promising potential of open-source models in the visual math reasoning domain. Moreover, there is a clear scaling trend observed in open-source models, indicating higher performance as model sizes increase. For example, Qwen2-VL boosts its score from 42.1% to 55.1% when scaling its parameter size from 7B to 72B, while InternVL2 sees an increase from 39.7% to 54.0%.

**Overall Results on Worst-case Accuracy.** Table 3 presents the worst-case accuracy of different models across 10 problem variants, revealing a significant decline in scores for all models. Notably, the highest-performing model, Claude-3.5, achieves a zero-shot score of only 35.3%, indicating current VLMs are not sufficiently robust to handle variations in context and images. The situation is even more concerning for open-source models: the best-performing model, Qwen2-VL-72B, achieves a score of 28.3%, while smaller models like Llava-v1.6-vicuna-13B score only 2.8%. Our evaluation results highlight the limited reasoning robustness of both open-source and closed-source models, underscoring the necessity for the community to address these limitations in future research.

**Fine-grained Results.** In Table 2 and Table 3, we present detailed results categorized by different question topics and difficulty levels. From a topical perspective, we observe that the Puzzle Test (PT) topic challenges both open-source and closed-source models. The top-performing closed-source model, GPT-4o, and the leading open-source model, InternVL2-76B, achieve average-case accuracies of 51.8% and 35.3%, respectively, while humans score 73.5%. Notably, all open-source models demonstrate poor performance (0.0%) on the worst-case accuracy metric, except InternVL2-76B (5.9%). Despite this gap, Table 2 shows that closed-source models such as Claude-3.5 can surpass human scores on specific topics like Algebra (AL) and Statistics (ST), which is promising. When considering difficulty levels, all models demonstrate a trend of decreasing average accuracy as the difficulty increases, as illustrated in Table 2. In contrast, human performance remains consistent across difficulty levels, indicating that current VLMs are still not adept at handling more difficult visual math problems compared with human capabilities.

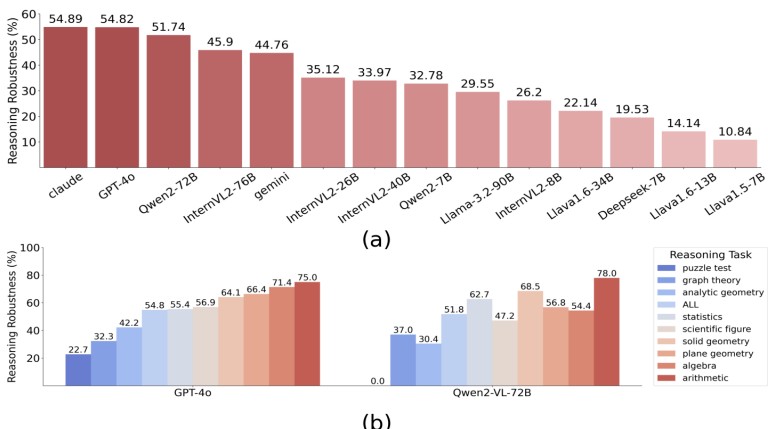

Figure 5: Comparing reasoning robustness across different (a) models and (b) topics.

| Model name | GPT-4o | Gemini | Qwen2-VL-72B | InternVL2-76B |
|---|---|---|---|---|
| Repetition Consistency (%) | 94.1 | 92.5 | 98.9 | 99.0 |

Table 4: The Repetition Consistency ($RC$) for different models over 5 repetitions.

**Reasoning Robustness.** We use the reasoning robustness ($RR$) metric, defined in Eq 2, to measure the robustness of VLMs by evaluating the relative performance consistency across question variants. We defer the detailed reasoning robustness results in Appendix H.3. Figure 5 (a) compares the $RR$ of all VLMs used in our experiments. Notably, Claude-3.5 and GPT-4o achieve the highest robustness among all tested models. Moreover, consistent with previous findings, closed-source models demonstrate greater robustness than open-source models, with reasoning robustness scaling with model size. **However, Qwen2-72B and InternVL2-76B outperform Gemini, highlighting the robustness limitations of even large models like Gemini.** In Figure 5 (b), we compare the reasoning robustness across different question topics for GPT-4o and Qwen2-VL-72B. The results show that the two VLMs are particularly robust in Arithmetic and Algebra question types, indicating their strong arithmetic calculation abilities, which are less affected by changes in visual conditions. However, GPT-4o still exhibits weaknesses in the Puzzle Test. Similarly, Qwen2-VL-72B shows shortcomings in both Puzzle Test and Analytic Geometry topics, achieving nearly 0% $RR$ and 30% $RR$, respectively. These weaknesses suggest directions for future improvement of these models.

**Repetition Consistency.** To ensure a robust analysis and account for the inherent randomness in model outputs, we calculate repetition consistency ($RC$) as defined in Eq 3. This metric evaluates the model's output confidence across multiple generations for the same question. Specifically, we produce five responses for 501 questions and then compute their consistency relative to the first response. The results, detailed in Table 4, reveal the consistent outputs of four closed-source and open-source models, with $RC$ values ranging from 92% to 99%. Compared with the low reasoning robustness scores, VLMs have much smaller consistency on different question variants. These findings reinforce our arguments that VLMs lack robustness in varying question conditions.

**Consistent Failure Cases.** An interesting phenomenon we observed is that some seed questions are solvable in certain variants but result in consistent failures in others (repetition consistency $RC = 1$ for 5 or 10 repetitions). The example in Figure 1 is a representative case: the question is easily solvable when the absolute value function at the origin, but any shifts tend to lead to consistent failures on GPT-4o. We extensively examined our dataset and counted the number of such instances. Specifically, GPT-4o, Gemini Pro 1.5, Qwen2-VL-72B, and InternVL2-76B exhibited 21.8%, 18.4%, 29.9%, and 28.3% of these types of questions, respectively, out of our 501 seed questions. These examples highlight the unreliability of VLMs on mathematical reasoning tasks.

### 4.3 QUALITY STUDY

**Qualitative Examples of GPT-4o.** In this section and Appendix G, we provide a few qualitative examples of leading VLMs' answers. Our analysis reveals that current VLMs can consistently produce incorrect responses to specific question variants while generating accurate answers to others. As illustrated in Figure 1, GPT-4o demonstrates the ability to provide correct responses in variant 7, showcasing accurate perception, question understanding, and reasoning ability. However, in variant 9, where the underlying required capabilities remain the same with only a slight shift in the image,

Seed Question 12: What is the period of this function? Answer the question with a floating-point number.

| Variant 1 ✓ | Variant 2 ✕ | Variant 3 ✕ | Variant 4 ✕ | Variant 5 ✕ |
|---|---|---|---|---|
| Answer: 6.283 | Answer: 6.283 | Answer: 6.283 | Answer: 6.283 | Answer: 6.283 |

Figure 7: Example of the Memorization Phenomenon: the generated variants of seed Question 12 and the corresponding responses from Claude 3.5 Sonnet. The model's response remains $2\pi$ with high probability, regardless of changes in the conditions depicted in the diagram.

GPT-4o fails to accurately interpret the function's position with a high degree of confidence and consistency. This discrepancy raises concerns about the reasoning robustness of current VLMs. For additional examples of GPT-4o and other models, please refer to the Appendix G.

**Memorization Phenomenon.** In our experiments, we observe a phenomenon where current VLMs tend to provide the same answer regardless of changing conditions, indicating memorization rather than reasoning based on generalized underlying principles. When we test variant questions that have the same structure but different parameters and images, the model frequently offers the same answer with high probability, ignoring the specific variations we introduced. Among the 171 questions incorrectly answered by Claude 3.5 Sonnet, this issue accounts for 4.1% of instances. A representative case is illustrated in Figure 7, where altering the period of a sinusoidal function (e.g., from $2\pi$ to $\pi$ or $4\pi$) does not affect the model's response, which consistently remains $2\pi$. The existence of this phenomenon highlights the models' lack of comprehensive problem analysis and their limited ability to generalize across different scenarios.

**Error Analysis.** We conducted an error analysis on Claude 3.5 Sonnet to identify potential failure modes on DYNAMATH. Specifically, we analyzed the 169 questions where Claude 3.5 Sonnet failed, examining the reasoning paths and final answers in detail. The statistical distribution of various error types is presented in Figure 6. We considered five types of errors: figure reading errors, reasoning errors, knowledge errors, calculation errors, and hallucination errors. Figure reading errors account for 33.1% of the total errors, despite Claude 3.5 Sonnet having specially reinforced perception capabilities. This indicates that there is still a considerable way to go for VLMs to accurately read and interpret data from images. Reasoning errors account for 26.6%, making them the second-largest cause of errors. This suggests that the model's

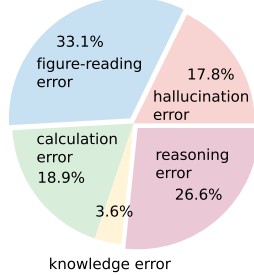

Figure 6: Error Analysis of Claude-3.5 Sonnet.

reasoning processes are still delicate and can be easily disrupted by minor changes in conditions and image input. Calculation errors, which constitute 18.9% of the errors, likely result from the significant computational challenge imposed by our randomly generated conditions without specially designed parameters, as opposed to simpler questions in prior work that are easier to compute. In addition, Hallucination errors make up 17.8%, showing that the model tends to fabricate non-existent information. More failure examples can be found in Appendix F.

## 5 CONCLUSION

In this work, we introduce DYNAMATH, a dynamic visual math benchmark designed to systematically analyze the robustness of mathematical reasoning capabilities in current leading vision-language models (VLMs). By employing program-based problem generation, we can create diverse variants by altering visual conditions in the seed problems. Our evaluation reveals that leading closed-source and open-source VLMs are sensitive to condition changes in question variants, despite their required underlying capabilities remaining the same. This raises significant concerns within the VLM community on mathematical reasoning tasks. Our detailed results and analysis not only identify the weak points of current VLMs but also shed light on the causes of their errors, thereby facilitating the development and evaluation of more robust VLMs in the future.

**Reproducibility Statement.** We have implemented several measures to ensure the reproducibility of our results. This includes providing detailed examples from our dataset, comprehensive descriptions of the prompts, and the hyperparameters used in our experiments. Additionally, our dataset is open-sourced for reproducibility.

**Acknowledgment** Huan Zhang is supported in part by the AI 2050 program at Schmidt Sciences (AI 2050 Early Career Fellowship). The authors thank OpenAI's researcher access program for providing part of the API credits used in our work.

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

## A    LIMITATIONS AND FUTURE WORK

Although our benchmark matches the difficulty levels of MATHVERSE and MATHVISTA, one limitation of our work is that the difficulty level is relatively limited compared to MATH-V (Wang et al., 2024a), due to the dynamic nature of the questions. Adapting very challenging questions into our program structures requires substantial human effort, which currently prevents us from curating a large number of complex visual math reasoning questions. In the future, we hope to leverage strong foundational models to aid in designing an automatic pipeline for dynamic math question design and generation.

Furthermore, the selection of seed questions can introduce unintended bias in DYNAMATH dataset. For instance, the most challenging question topic for VLMs, the Puzzle test, only dominates 3.4% of the whole dataset. It remains an open problem to study the bias in open-source datasets and requires further efforts. Regarding the variation generation process, we have identified a limitation: we currently consider only individual types of variants, such as Numerical Value Variants or Function Type Variants, for each seed question. However, in many cases, it is possible to combine different types of variants, such as Color Variants and Numerical Value Variants. We will explore the integration of different variant types to further investigate the reasoning robustness of VLMs.

**Scalability of DYNAMATH**    The current design of DYNAMATH relies heavily on human effort to curate high-quality seed questions. However, it is important to scale up the design process of DynaMATH for constructing more comprehensive and challenging benchmarks. Below, we outline the primary challenges and discuss potential solutions:

A key challenge in scaling DYNAMATH is incorporating dynamic visual elements for each question. Unlike text-only benchmarks, our dataset includes an image for every problem with different variants (e.g., graphs, geometric shapes, function plots, real-life content). This requires careful design of the drawing program, adding significant manual effort, especially in quality control and verification, which complicates full automation.

A promising solution is to leverage LLMs to automate the generation of dynamic benchmarks. LLMs have shown proficiency in generating text-based problems and writing code (Mirzadeh et al., 2024). It is possible to break down benchmark topics and subtopics, prompting the LLM to generate diverse problem sets and corresponding Python programs for visual elements. However, the generated problems should be dynamic, with parameterizable Python code to produce multiple image variants. To this end, DYNAMATH is a valuable benchmark since our seed questions can serve as high-quality human demonstrations to guide the LLMs for this task. This LLM-assisted approach could significantly reduce manual effort. However, some human intervention will still be necessary to ensure the selection of correct and high-quality samples from LLMs.

While we have to leave the LLM-assisted dynamic benchmark generation as a future work, DYNA-MATH can serve as a good baseline that is completely crafted by human beings, and future work on automated dynamic benchmark generation may compare to DYNAMATH in terms of diversity and quality.

**Future Work**    Moving forward, an intriguing approach to enhance VLM robustness involves leveraging adversarial training (Zhou et al., 2024; Schlarmann et al., 2024) on DYNAMATH, or utilizing reinforcement learning from human feedback (Ouyang et al., 2022; Sun, 2023; Rafailov et al., 2024; Yang et al., 2024b) with fine-grained process rewards (Uesato et al., 2022; Wang et al., 2024b; Luo et al., 2024), or more robust rewards (Yang et al., 2024a; Zhang et al., 2024c). While prior successes in robust machine learning (Silva & Najafirad, 2020; Zhang et al., 2020; Yang et al., 2022) and trustworthy LLMs (Sun et al., 2024) offer valuable insights, adapting these methods to VLMs in a cost-effective way remains an open challenge and an area of potential exploration.

## B    VARIATION TYPES OF DYNAMATH

DYNAMATH introduces several types of variations based on the seed questions. In Figure 8, we illustrate six distinct types of variations. This diversity allows our dataset to effectively evaluate the visual robustness of VLMs.

$$A = \begin{bmatrix} -4.0 & 4.6 & 9.7 \\ -1.0 & 8.3 & -3.3 \\ -0.3 & -7.4 & -6.2 \end{bmatrix} \quad A = \begin{bmatrix} -1.9 & 9.5 & -7.0 \\ -7.2 & -5.3 & 4.2 \\ -0.7 & -6.0 & -0.4 \end{bmatrix}$$

$$A = \begin{bmatrix} -0.2 & -0.2 & -6.9 \\ 8.2 & 2.9 & -5.5 \\ -0.4 & 8.7 & -8.5 \end{bmatrix} \quad A = \begin{bmatrix} -5.1 & 1.1 & -4.0 \\ -5.2 & 7.8 & 2.6 \\ 3.5 & -7.2 & 0.0 \end{bmatrix}$$

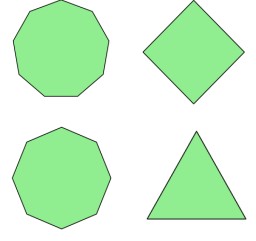

| | |
|---|---|
| **(a) Numerical Value Variants** | **(b) Geometric Transformations** |

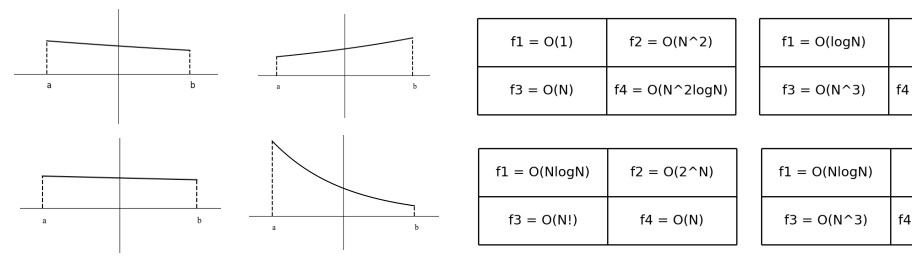

| | |
|---|---|
| **(c) Function Type Variants** | **(d) Symbolic Substitution** |

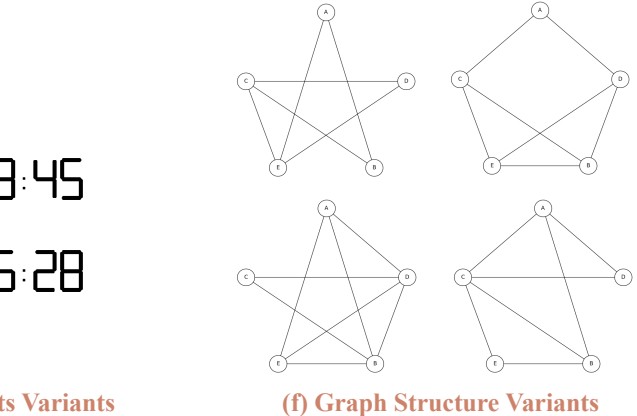

| | |
|---|---|
| **(e) Real-life Contexts Variants** | **(f) Graph Structure Variants** |

Figure 8: Variation types considered in our DYNAMATH benchmark

## C  DETAILED EXPERIMENT SETUP

In this section, we provide more details about our experiment designs.

### C.1  PROMPTS FOR RESPONSE GENERATION

In our experiments, we prompt the VLMs to generate responses to different types of questions, such as multiple choice, float, and text types. The prompts used for these question types are shown in Table 5.

### C.2  PROMPTS FOR ANSWER EXTRACTION

To simplify the answer extraction process, we use the following prompts to extract the answer and reload it in JSON format, which can be easily used for template matching with ground truth answers:

| Answer type | prompt |
|---|---|
| multiple choice | If the problem is a multiple choice problem, just provide the corresponding choice option, such as 'A', 'B', 'C', or 'D'. |
| float | If the answer is a numerical value, format it as a three-digit floating-point number. |
| text | Please answer the question in the following form: (specific requirement in question). |

Table 5: The prompt for different questions and answer types in answer generation.

---

**Response Instruction**

Please provide an answer to the question outlined above. Your response should adhere to the following JSON format, which includes two keys: 'solution' and 'short answer'. The 'solution' key can contain reasoning steps needed to solve the question. The 'short answer' key should only provide a concise response.
Example of expected JSON response format:

```
{
  "solution": "[Detailed step-by-step explanation]",
  "short answer": "[Concise Answer]"
}
```

---

Closed-source models can consistently extract answers and regenerate them in JSON format. In contrast, some open-source models, particularly those with relatively low parameter sizes, struggle to output JSON format reliably. We have identified patterns in their inconsistent responses, which deviate from the expected code scripts. By supplementing these scripts, we efficiently check, annotate, and extract informal answers.

## C.3    PROMPTS FOR FEW-SHOT CHAIN OF THOUGHT

Below we provide the few-shot CoT prompt for plane geometry topic. More demonstration examples for each topic can be found in Appendix E

---

**Few-shot CoT Prompt**

Please answer the question based on the last image. To clearly explain the task, we provide the following example:

For image 1: Please answer the question and provide the correct option letter, e.g., A, B, C, D, at the end. Question: As shown in the figure, $AB$ is the diameter of the semicircle, point $O$ is the center of the circle, point $C$ is a point on the extended line of $AB$, and $CD$ is tangent to the semicircle at point $D$. If $AB = 6.0$, $CD = 4.0$, then the value of $\sin(\angle C)$ is ()

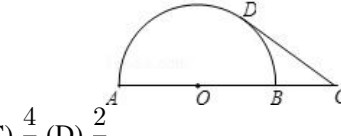

Choices: (A) $\dfrac{3}{4}$ (B) $\dfrac{3}{5}$ (C) $\dfrac{4}{5}$ (D) $\dfrac{2}{3}$

Answer: (B)

Let's think this step by step:

To solve this problem, we can make use of the properties of the circle and the tangent line. Here's a step-by-step breakdown of the approach:

1. **Recognize the Geometry:** - $AB$ is the diameter of the semicircle, so the length of $AB$ is 6. This makes the radius of the circle $AO = BO = 3$. - $CD$ is tangent to the semicircle at point $D$. Tangents to a circle are perpendicular to the radius at the point of tangency, meaning $OD \perp CD$.

2. **Set Up the Triangle:** - We are asked to find $\sin(\angle C)$. The triangle we should focus on is triangle $ODC$. - By the properties of a tangent, $\angle ODC = 90°$.

3. **Apply Pythagoras' Theorem:** - In the right triangle $ODC$, we have the lengths $OD = 3$ (since it's the radius of the circle), and $CD = 4$ (given). - To find $OC$, use the Pythagorean theorem:
$$OC^2 = OD^2 + CD^2$$
$$OC^2 = 3^2 + 4^2 = 9 + 16 = 25$$
$$OC = 5$$

4. **Find** $\sin(\angle C)$**:** - From triangle $ODC$, $\sin(\angle C) = \frac{OD}{OC}$.
$$\sin(\angle C) = \frac{3}{5}$$

Thus, the correct option is $B$.

For image 2: ...

For image 3: ...

Now please answer the following question based on the last image: Find the perimeter of the orange triangle. Please answer in a floating-point number.

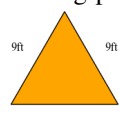

---

## C.4 MODEL HYPERPARAMETERS

We set all parameters except temperature to their default values. We set temperature = 0 for closed-source models and open-source models to reduce the randomness in the model generation. Table 6 displays the parameters we used for generation in VLMs.

Table 6: Hyperparameters for various VLMs.

| Model | Hyperparameters |
| --- | --- |
| GPT-4o | model = `gpt-4o-0806`, temperature = 0.0, max_tokens = 4096 |
| Claude-3.5 | model = `claude-3-5-sonnet-20240620`, temperature = 0.0, max_tokens = 1024 |
| Gemini Pro 1.5 | model = `gemini-1.5-pro`, temperature = 0.0, max_tokens = 8192 |
| Qwen2-VL-72B | model = `Qwen/Qwen2-VL-72B-Instruct`, temperature = 0.0, max_tokens = 2048 |
| QWen2-VL-7B | model = `Qwen/Qwen2-VL-7B-Instruct`, temperature = 0.0, max_tokens = 2048 |
| InternVL2-76B | model = `OpenGVLab/InternVL2-Llama3-76B`, temperature = 0.0, max_tokens = 1024 |
| InternVL2-40B | model = `OpenGVLab/InternVL2-40B`, temperature = 0.0, max_tokens = 1024 |
| InternVL2-26B | model = `OpenGVLab/InternVL2-26B`, temperature = 0.0, max_tokens = 1024 |
| InternVL2-8B | model = `OpenGVLab/InternVL2-8B`, temperature = 0.0, max_tokens = 1024 |
| Deepseek-VL-7B-chat | model = `deepseek-ai/deepseek-vl-7b-chat`, temperature = 0.0, max_tokens = 1024 |
| Llama-3.2-90B | model = `meta-llama/Llama-3.2-90B-Vision-Instruct`, temperature = 0.0, max_tokens = 1024 |
| Llava-v1.6-34B | model = `liuhaotian/llava-v1.6-34b`, temperature = 0.0, max_tokens = 1024 |
| Llava-v1.6-vicuna-13B | model = `liuhaotian/llava-v1.6-vicuna-13b`, temperature = 0.0, max_tokens = 1024 |
| Llava-v1.5-7B | model = `liuhaotian/llava-v1.5-7b`, temperature = 0.0, max_tokens = 1024 |

# D    VARIANT EXAMPLES FOR DIFFERENT TOPICS IN DYNAMATH

In this section, we show sample problems in DYNAMATH for different topics including multiple variants, including Solid Geometry (SG), Puzzle Tests (PT), Arithmetic (AR), Scientific Figures (SF), Graph Theory (GT), Algebra (AL), Plane Geometry (PG), Analytic Geometry (AG), and Statistics (ST).

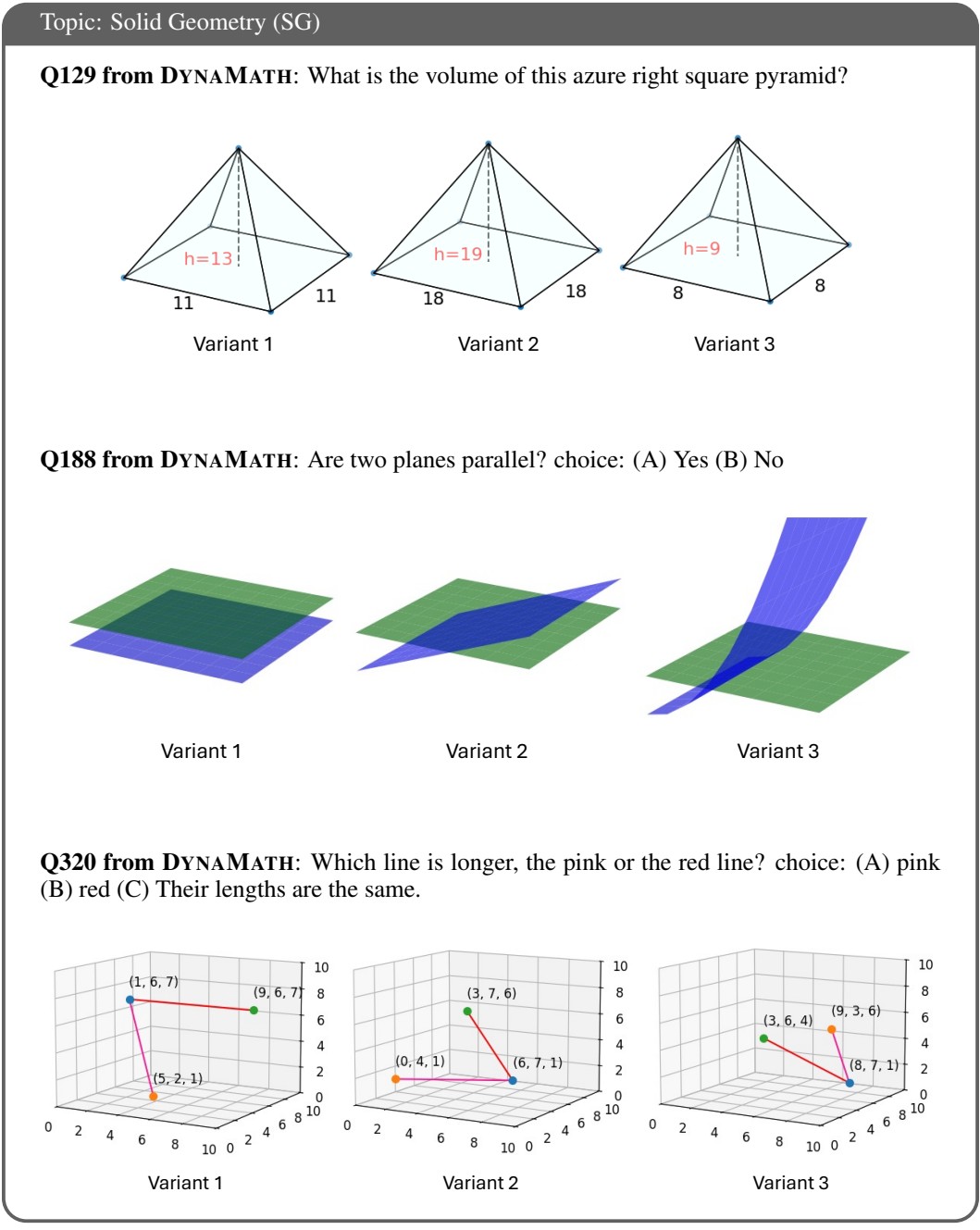

Topic: Puzzle test (PT)

**Q115 from DYNAMATH**: The sum of the three numbers on each of the two lines of the cross is 76. Find the number in the center.

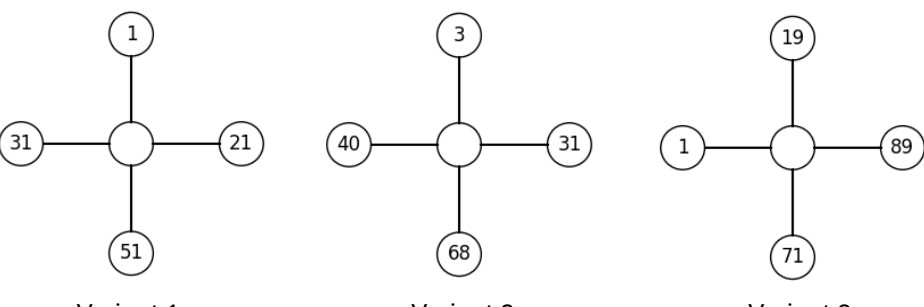

Variant 1      Variant 2      Variant 3

**Q282 from DYNAMATH**: Fill in the white spaces to make the equations work. choice: (A) 13, 25, 5, and 12 (B) 25, 5, 12, and 12 (C) 13, 4, 25, 13.

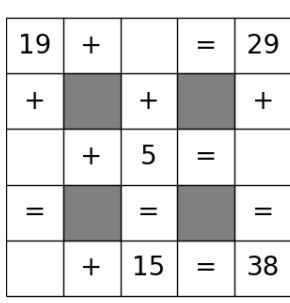
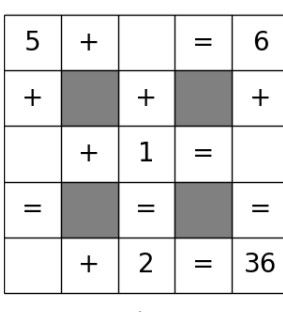
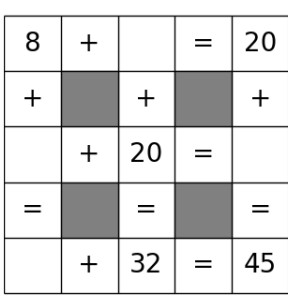

Variant 1      Variant 2      Variant 3

**Q284 from DYNAMATH**: Find the missing value.

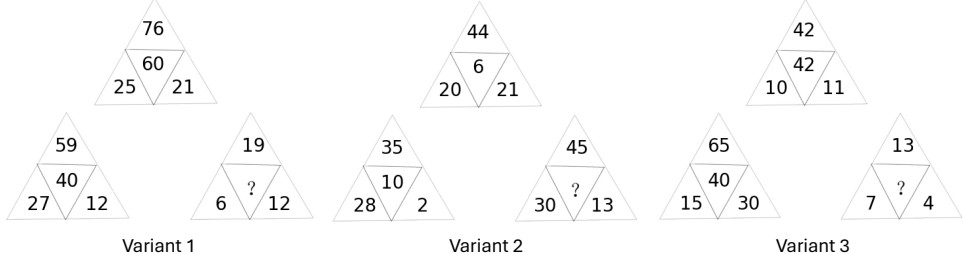

Variant 1      Variant 2      Variant 3

Topic: Arithmetic (AR)

**Q7 from DYNAMATH**: In the addition sum to the right, three digits have been replaced with star. What is the value of star?

```
    1 * 8             1 * 7             1 * 3
+ 1 * 0           + 1 * 2           + 1 * 1
+ 1 * 9           + 1 * 7           + 1 * 5
-----------       -----------       -----------
    4 6 7             5 2 6             5 4 9
```

Variant 1               Variant 2               Variant 3

**Q25 from DYNAMATH**: What is the missing computed symbol? Choices: (A) + (B) - (C) * (D) /

4 + 8 - 4 __ 4 = 12     8 + 3 - 4 __ 10 = -3     3 + 9 - 2 __ 6 = 4

Variant 1               Variant 2               Variant 3

**Q316 from DYNAMATH**: According to the boarding pass, how long is the flight time of this airplane? Answer the question using the total number of minutes.

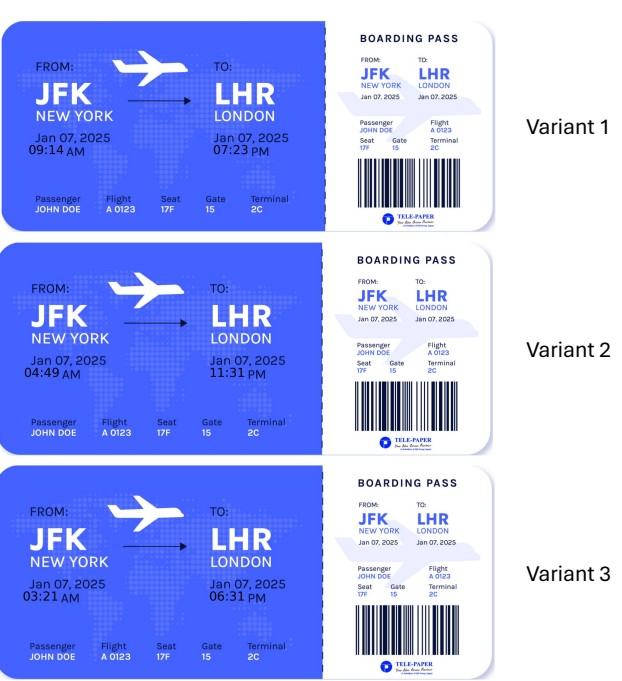

Topic: Scientific figure (SF)

**Q323 from DYNAMATH**: Two containers of the same gas (ideal) have these masses and temperatures Which box has atoms with the largest average thermal energy? choice: (A) A (B) B (C) Their average thermal energy is the same.

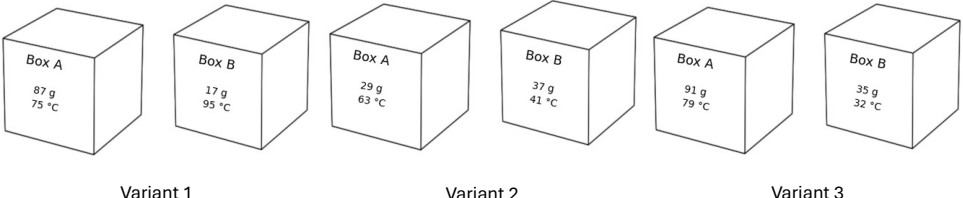

Variant 1                                Variant 2                                Variant 3

**Q325 from DYNAMATH**: Three equally spaced identical long straight wires carry different currents. In which direction will the middle wire try to move when the currents are switched on? choice: (A) to the left (B) to the right (C) stay the same

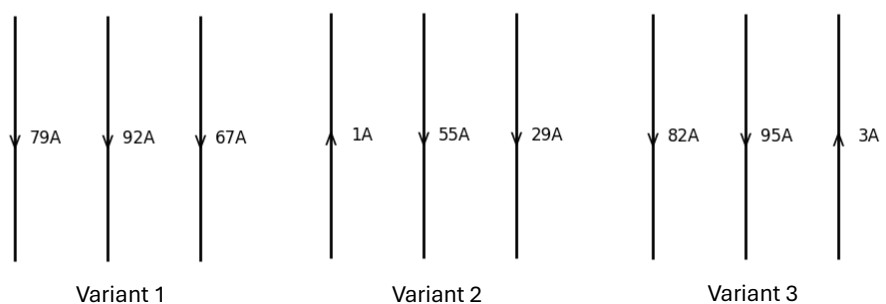

Variant 1                                Variant 2                                Variant 3

**Q331 from DYNAMATH**: The graph shows the force on an object of mass M as a function of time. For the time interval 0 to 10 s, what is the total change in the momentum of the object?

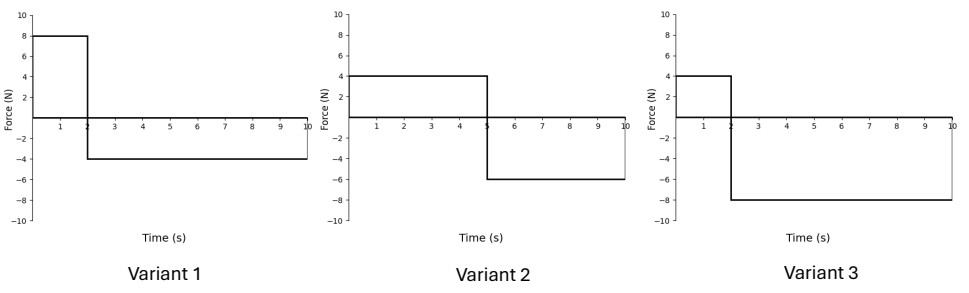

Variant 1                                Variant 2                                Variant 3

Topic: Graph theory (GT)

**Q42 from DYNAMATH**: Is the graph shown connected? choice: (A) Yes (B) No

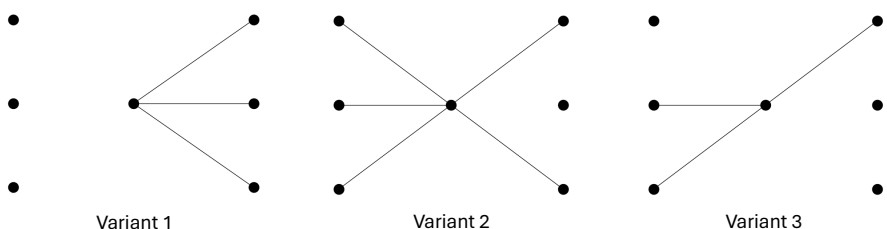

**Q137 from DYNAMATH**: What is the first edge added to the MST when running Kruskal's Algorithm? In the case of a tie, choose the edge which comes first in alphabetical order i.e. if you had to choose between AS and AE, then you would choose AE first.

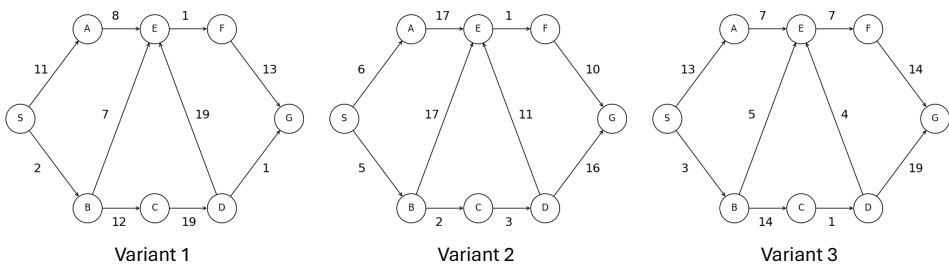

**Q259 from DYNAMATH**: The tree shown in image reserves an expression. Calculate this expression and output the result.

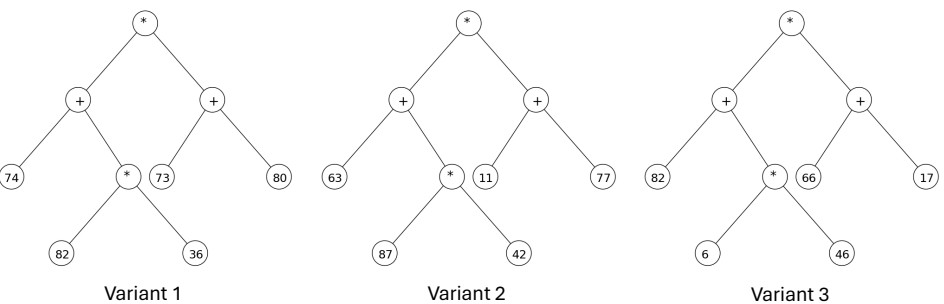

Topic: Algebra (AL)

**Q305 from DYNAMATH**: The store has 4 combinations of candies. Each candy type has the same price. Find the price of the fourth combination.

| | | |
|---|---|---|
| 1. ★ ★ ★ ★ ★   $15 | 1. ★ ★ ★ ★ ★   $20 | 1. ★ ★ ★ ★ ★   $10 |
| 2. ★ ■ ★ ■ ★   $47 | 2. ★ ■ ★ ■ ★   $44 | 2. ★ ■ ★ ■ ★   $10 |
| 3. ▲ ■ ▲ ■ ▲   $71 | 3. ▲ ■ ▲ ■ ▲   $77 | 3. ▲ ■ ▲ ■ ▲   $49 |
| 4. ★ ■ ■ ▲ ★   $? | 4. ★ ■ ■ ▲ ★   $? | 4. ★ ■ ■ ▲ ★   $? |
| Variant 1 | Variant 2 | Variant 3 |

**Q351 from DYNAMATH**: Which function has the highest order or growth? choice: (A) f1 (B) f2 (C) f3 (D) f4

| | | | | |
|---|---|---|---|---|
| f1 = O(N^2logN) | f2 = O(1) | f1 = O(2^N) | f2 = O(N^3) | f1 = O(N^2logN) | f2 = O(N) |
| f3 = O(N!) | f4 = O(2^N) | f3 = O(logN) | f4 = O(1) | f3 = O(2^N) | f4 = O(logN) |

| Variant 1 | Variant 2 | Variant 3 |
|---|---|---|

**Q465 from DYNAMATH**: 210 customers were surveyed about their product preferences. The results are displayed in the Venn diagram below. How many more customers prefer only Non-Organic products than only Organic ones?

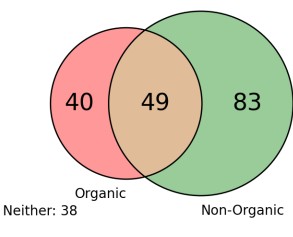

Neither: 38
Organic   Non-Organic
Variant 1

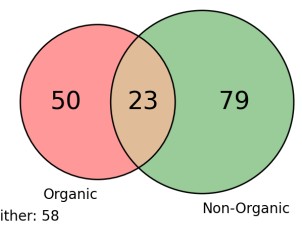

Neither: 12
Organic   Non-Organic
Variant 2

Neither: 58
Organic   Non-Organic
Variant 3

Topic: Plane geometry (PG)

**Q28 from DYNAMATH**: The two rectangles shown in the picture have the same area. what is the ratio $x : y$.

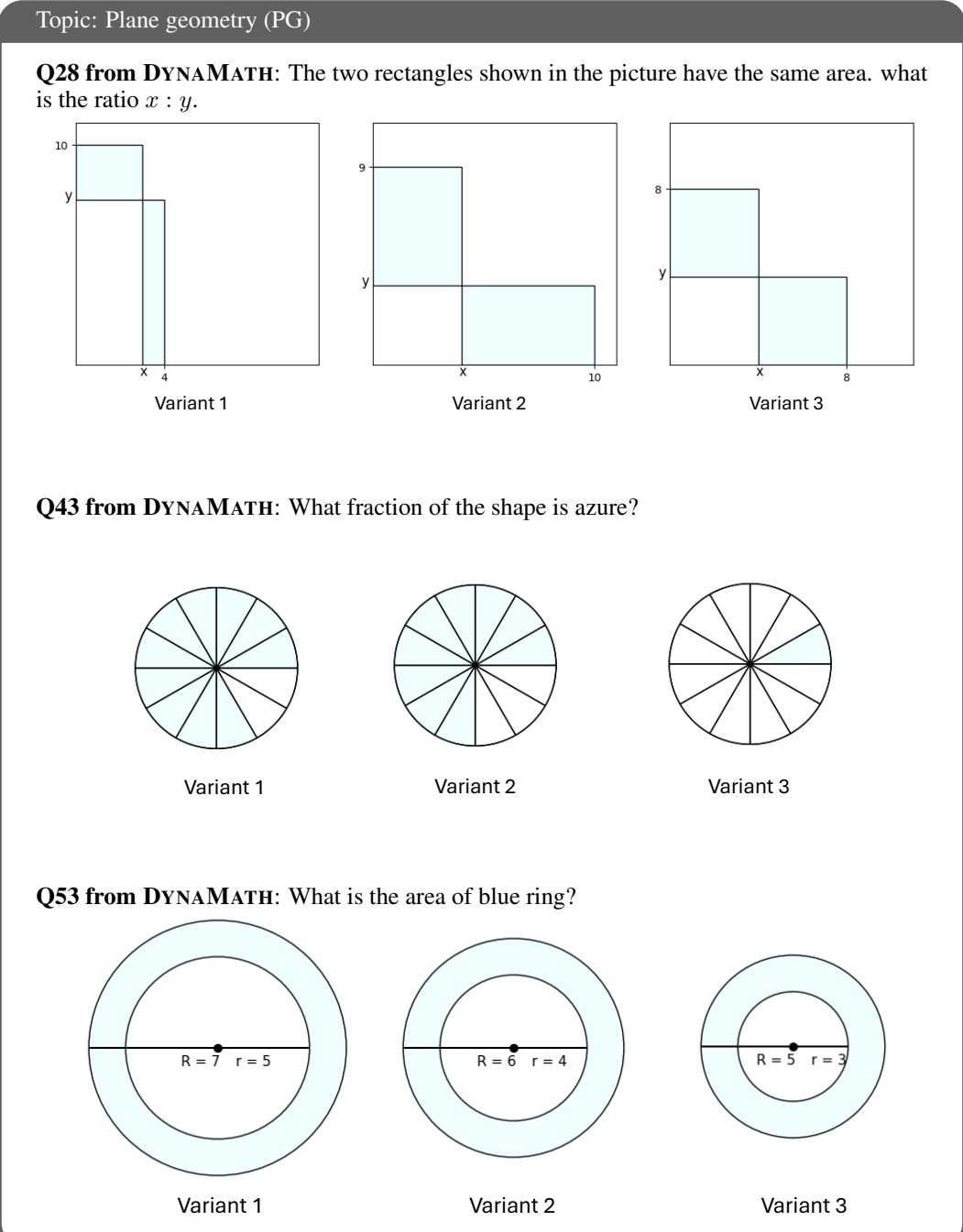

Variant 1            Variant 2            Variant 3

**Q43 from DYNAMATH**: What fraction of the shape is azure?

Variant 1            Variant 2            Variant 3

**Q53 from DYNAMATH**: What is the area of blue ring?

Variant 1            Variant 2            Variant 3

Topic: Analytic geometry (AG)

**Q68 from DYNAMATH**: What is the green curve? choice: (A) a parabola (B) a line (C) a logarithmic function (D) a trigonometric function.

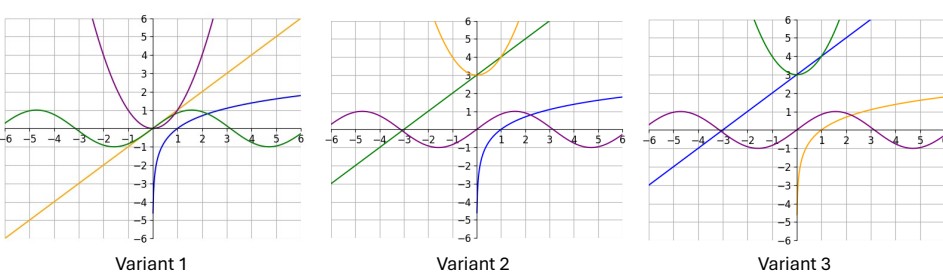

Variant 1    Variant 2    Variant 3

**Q87 from DYNAMATH**: What is the limit of the function as x approaches 1 from the left side?

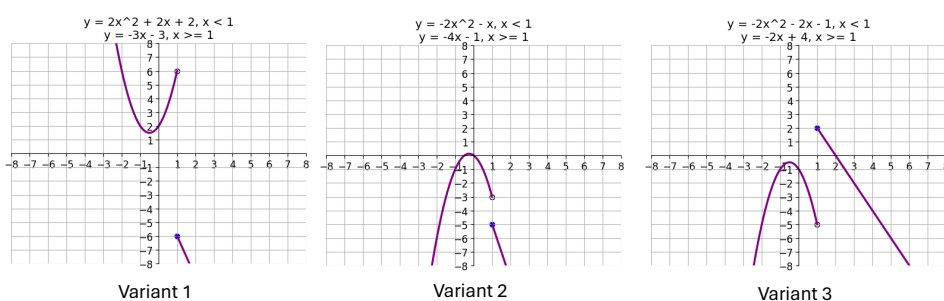

Variant 1    Variant 2    Variant 3

**Q111 from DYNAMATH**: The image shows the derivative of $f(x)$. Where is the local max of $f(x)$ at?

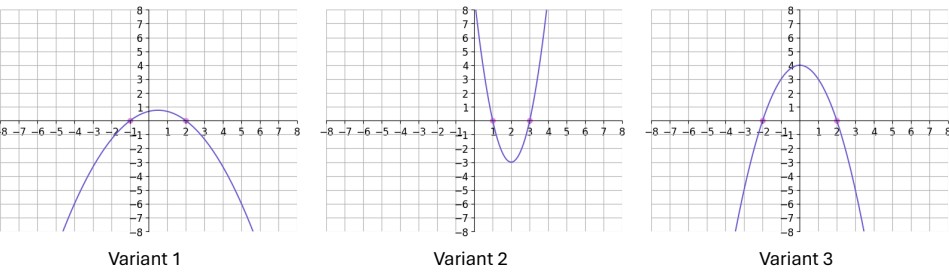

Variant 1    Variant 2    Variant 3

Topic: Statistics (ST)

**Q72 from DYNAMATH**: According to the markov chain shown in the image, what is the probability of the event 'A to B'?

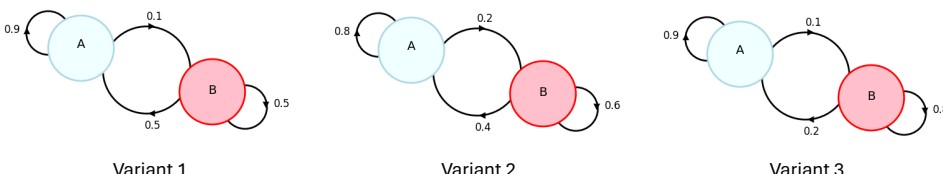

Variant 1          Variant 2          Variant 3

**Q161 from DYNAMATH**: On which number is the spinner more likely to land?

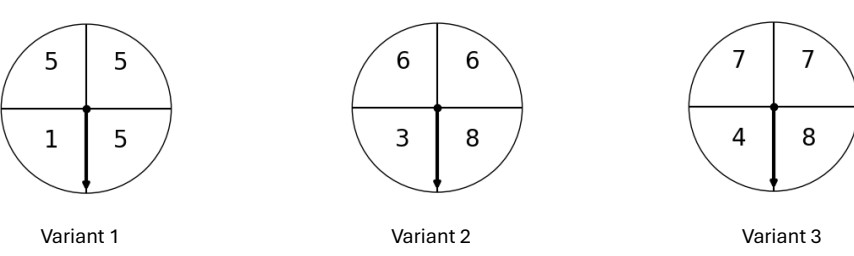

Variant 1          Variant 2          Variant 3

**Q447 from DYNAMATH**: The Gantt chart below represents different tasks. Which task starts the earliest? Choices: (A) Task A (B) Task B (C) Task C (D) Task D (E) Task E

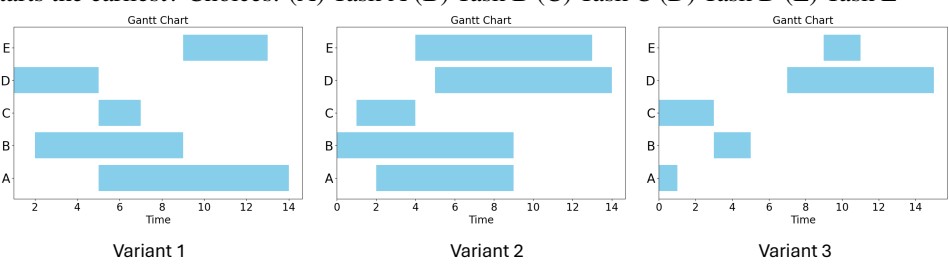

Variant 1          Variant 2          Variant 3

# E  MORE DEMONSTRATION EXAMPLES FOR FEW-SHOT EXPERIMENT

---

**Demonstration Examples for Topic Algebra**

The two markers with a question mark have the same number. Which number do you have to put instead of the question mark so that the calculation is correct?

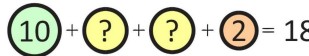

Answer: 3

Let's think this step by step: To solve the problem: The equation is: $10 + ? + ? + 2 = 18$
Where both question marks represent the same number. Let's denote the number as $x$. The equation becomes: $10 + x + x + 2 = 18$
Combine like terms: $12 + 2x = 18$.
Now, solve for $x$: $2x = 18 - 12$, hence $2x = 6$, and therefore $x = 3$.
So, each question mark should be replaced with 3 to satisfy the equation.

---

**Demonstration Examples for Topic Arithmetic**

How many plums (see the picture) weigh as much as an apple?

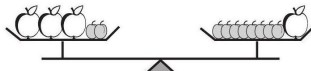

Answer: 3

Let's think this step by step:
Since the scale is balanced, the total weight on the left side equals the total weight on the right side. 1. Let the weight of one apple be $A$, and the weight of one plum be $P$.
2. The total weight on the left side of the scale: $3A + 2P$.
3. The total weight on the right side of the scale: $A + 8P$.
Since the scale is balanced, we can set the two expressions equal to each other: $3A + 2P = A + 8P$.
### Solving for $A$ in terms of $P$:
1. Subtract $A$ from both sides: $2A + 2P = 8P$.
2. Subtract $2P$ from both sides: $2A = 6P$.
3. Divide both sides by 2: $A = 3P$.
### Final Answer: One apple weighs as much as 3 plums.

Demonstration Examples for Scientific Figure

If the population of the red squirrels increased, what would happen to the amount of pine available?
(A) The amount of pine available would remain the same.
(B) The amount of pine available would decrease.
(C) The amount of pine available would double.
(D) The amount of pine available would increase.

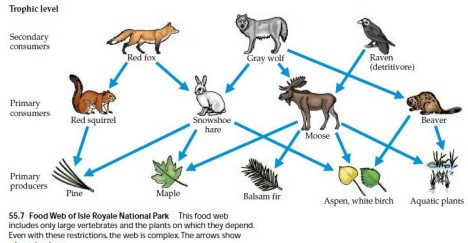

**55.7 Food Web of Isle Royale National Park** This food web includes only large vertebrates and the plants on which they depend. Even with these restrictions, the web is complex. The arrows show who eats whom.

Answer: (B)

Let's think this step by step: In the food web of Isle Royale National Park, the red squirrels are primary consumers that feed on pine, which is a primary producer.
If the population of red squirrels increases, the consumption of pine would increase because more squirrels would be feeding on it. As a result, the amount of pine available would decrease.
**Conclusion**: If the population of red squirrels increased, the amount of pine available would decrease. Thus, the correct option is $B$.

---

**Demonstration Examples for Topic Puzzle Test**

Please determine the total weight in the fourth image.

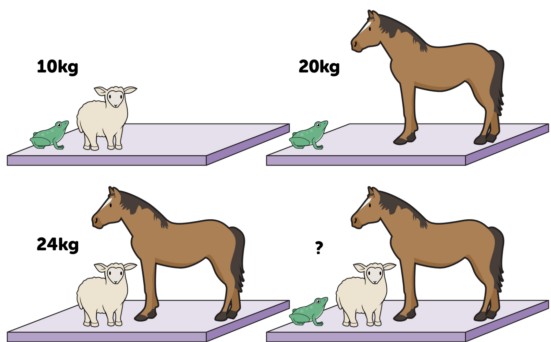

Answer: 27

To solve the puzzle and determine the total weight in the fourth image, let's break it down based on the given weights:

1. **First image**: A frog and a sheep together weigh 10 kg.
2. **Second image**: A horse and a frog together weigh 20 kg.
3. **Third image**: A horse and a sheep together weigh 24 kg.

We can now set up three equations to represent these situations:
- Frog + Sheep = 10 kg (Equation 1)
- Horse + Frog = 20 kg (Equation 2)
- Horse + Sheep = 24 kg (Equation 3)

Solving the system of equations:
1. From Equation 2, we can express the weight of the horse:
- Horse = 20 kg - Frog (from Equation 2).
2. Substitute Horse in Equation 3:
- (20 kg - Frog) + Sheep = 24 kg.
- 20 kg - Frog + Sheep = 24 kg.
- Sheep - Frog = 4 kg (Equation 4).
3. Now use Equation 1 (Frog + Sheep = 10 kg) and Equation 4 (Sheep - Frog = 4 kg):
- Adding these two equations: (Frog + Sheep) + (Sheep - Frog) = 10 kg + 4 kg, gives:
$2 \times$ Sheep = 14 kg, so Sheep = 7 kg.
4. Substitute the value of Sheep (7 kg) into Equation 1:
- Frog + 7 kg = 10 kg, so Frog = 3 kg.
5. Finally, substitute the value of Frog (3 kg) into Equation 2:
- Horse + 3 kg = 20 kg, so Horse = 17 kg.
Total weight in the fourth image:
- The fourth image has the horse, the sheep, and the frog together.
- The total weight is:
Horse (17 kg) + Sheep (7 kg) + Frog (3 kg) = 27 kg.

Thus, the total weight in the fourth image is 27 kg.

---

# F   ERROR TYPE EXAMPLES

In this section, we present various examples of error types encountered in Claude 3.5, including figure reading error, calculation error, reasoning error, hallucination error, and knowledge error.

**Figure Reading Error**

**Q96 from DYNAMATH**: What is the distance between the x-intercept and y-intercept of blue line?

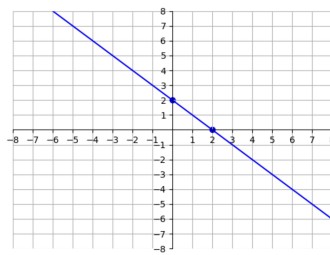

**Claude-3.5 Answer**:
1. Identify the $x$-intercept and y-intercept: The $x$-intercept is where the line crosses the $x$-axis ($y = 0$). From the graph, this appears to be at $(3, 0)$. The $y$-intercept is where the line crosses the $y$-axis ($x = 0$). From the graph, this appears to be at $(0, 4)$.
2. Calculate the distance using the distance formula: distance = $\sqrt{(x_2 - x_1)^2 + (y_2 - y_1)^2}$ where $(x_1, y_1)$ is $(3, 0)$ and $(x_2, y_2)$ is $(0, 4)$.
3. distance = $\sqrt{(0 - 3)^2 + (4 - 0)^2} = 5$. Therefore, the distance between the $x$-intercept and $y$-intercept is 5 units.

**Ground Truth**: Distance = $\sqrt{4} \approx 2.828$.

**Error Analysis:** The model misreading the coordinate of the points and gives the wrong answer.

**Figure Reading Error 2**

**Q185 from DYNAMATH**: Based on the measurement results shown in the diagram, this blue angle is ( ).

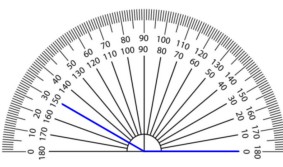

**Claude-3.5 Answer**: To determine the angle represented by the blue lines in the protractor image:
1. Observe that the protractor is a 180-degree semicircle.
2. Note that one blue line is at the 0-degree mark.
3. Locate where the other blue line intersects the protractor scale.
4. This intersection occurs at approximately 25 degrees.
5. Since the protractor measures angles from 0 to 180 degrees, the angle formed by the blue lines is directly read as 25 degrees.
Therefore, the measurement of the blue angle is approximately 25 degrees.

**Ground Truth**: 150 degrees.

**Error Analysis:** The model fail to distinguish between two set of the numbers for degree reading. In addition, even for outer degree numbers, the model gives an inaccurate number reading.

---

**Calculation Error**

**Q108 from DYNAMATH**: The function f(x) is a cubic function. What is the value of f(x) when x = 4?

| x | f(x) |
|---|------|
| 0 | 3 |
| 1 | 19 |
| 2 | 63 |
| 3 | 159 |
| 4 | ? |

**Claude-3.5 Answer**:
1. First, let's recognize that for a cubic function $f(x) = ax^3 + bx^2 + cx + d$, we need to find the values of $a$, $b$, $c$, and $d$.
2. We can use the given points to set up a system of equations: $f(0) = d = 3$,
$f(1) = a + b + c + 3 = 19$,
$f(2) = 8a + 4b + 2c + 3 = 63$,
$f(3) = 27a + 9b + 3c + 3 = 159$.
3. From the first equation, we know $d = 3$.
4. Subtracting equations:
$f(1) - f(0) : a + b + c = 16$
$f(2) - f(1) : 7a + 3b + c = 44$
$f(3) - f(2) : 19a + 5b + c = 96$
5. Solving this system, we get: $a = 4$, $b = 7$, $c = 5$, $d = 3$
6. So, our function is $f(x) = 4x^3 + 7x^2 + 5x + 3$
7. Now, we can calculate $f(4)$: $f(4) = 4(4^3) + 7(4^2) + 5(4) + 3 = 256 + 112 + 20 + 3 = 391$
Therefore, $f(4) = 391$.

**Ground Truth**: Let $f(x) = ax^3 + bx^2 + cx + d$, using the number given in the table, one can get: $a = 4$, $b = 2$, $c = 10$, $d = 3$. So, our function is $f(x) = 4x^3 + 2x^2 + 10x + 3$ and $f(4) = 331$.

**Error Analysis:** The model fails to solve the system of equations for value $b$ and $c$.

---

**Reasoning Error 1**

**Q146 from DYNAMATH**: Find the missing number.

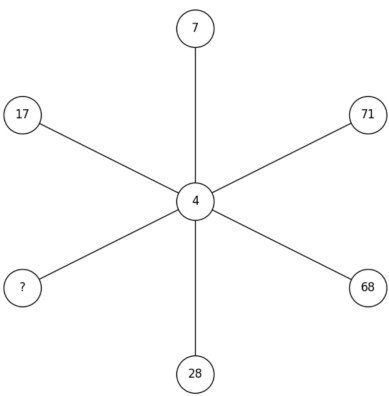

**Claude-3.5 Answer**:
1. The diagram shows a central number 4 with six numbers radiating from it.
2. The visible numbers are 7, 17, 28, 68, and 71.
3. Let's check if there's a consistent mathematical relationship between 4 and these numbers: 4 x 7 = 28 (one of the given numbers) 4 x 17 = 68 (another given number)
4. This suggests that each number is either a factor or multiple of 4.
5. 71 doesn't fit this pattern directly, but 71 + 4 = 75, which is 3 x 25.
6. Following this pattern, the missing number should also be related to 4 by multiplication or addition.
7. A number that fits this pattern and completes the sequence is 13, because: 4 x 13 = 52, and 52 + 4 = 56, which is 4 x 14.
8. Therefore, the missing number is likely 13.

**Ground Truth**: $71 \times 4 = 284$.

**Error Analysis:** The first 4 steps from the model are correct, and the model can observe that each number is either a factor or multiple of 4 correctly. However, step 5-8 is not correct and the reasoning to get its final answer 13 is not correct.

---

### Reasoning Error 2

**Q34 from DYNAMATH**: What is the area of this parallelogram?

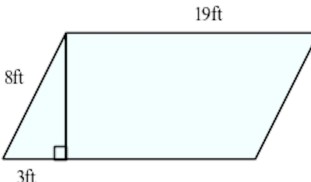

**Claude-3.5 Answer**: To find the area of a parallelogram, we use the formula: Area = base × height. In this case:
1. The base is given as 19 ft.
2. The height is the perpendicular distance between the base and the opposite side, which is 8 ft.
Area = 19 ft × 8 ft = 152 sq ft.

**Ground Truth**: To find the area of a parallelogram, we first need to compute the height $h = \sqrt{8^2 - 3^2} \approx 7.416$. Then the area of the parallelogram is $19 \times 7.416 = 140.904$ sq ft.

**Error Analysis:** The model fail to reason that we need to compute the height using the given numbers in the plot.

---

### Hallucination Error 1

**Q146 from DYNAMATH**: Find the area of the shaded part.

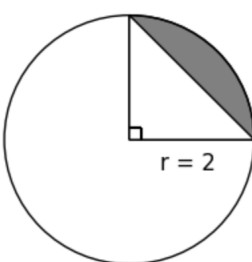

**Claude-3.5 Answer**:
1. The circle has a radius r = 2.
2. The shaded area is a sector of the circle minus a right triangle.
3. The sector angle is 1/8 of the full circle (45°).
4. Area of the sector $= (1/8)\pi r^2 = 1.5708$. Area of the triangle $= (1/2)r^2 = (1/2)2^2 = 2$.
6. Shaded area = Area of sector - Area of triangle $= 1.5708 - 2 = -0.42927$.
5. Taking the absolute value: $|-0.42927| = 0.4292$.

**Ground Truth**: Area of the shaded part $= (1/4)\pi r^2 - (1/2)r^2 = 1.1416$.

**Error Analysis:** The model hallucinates the sector angle to be $45°$, which should be $90°$. In addition, instead of realizing that the area should be a positive number, the model take the absolute value of a negative number and return it as the answer.

---

Hallucination Error 2

**Q120 from DYNAMATH**: Is this graph a cyclic graph? choice: (A) Yes (B) No

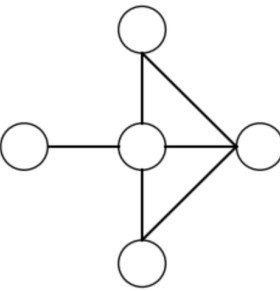

**Claude-3.5 Answer**:
To determine if this graph is cyclic, we need to check if there's a path that starts and ends at the same node without repeating any edges. Looking at the graph:
1. It has 5 nodes connected in a star-like pattern.
2. The central node is connected to all other nodes.
3. There are no connections between the outer nodes.
4. To form a cycle, we would need to be able to start at one node, traverse through others, and return to the starting node without repeating edges.
5. In this graph, any path that tries to return to a starting node would have to go through the central node twice, which means repeating an edge.
6. Therefore, there is no cycle in this graph.
Since there is no cycle, this graph is not cyclic. The final answer is (B) No.

**Ground Truth**: Since there is a path that starts and ends at the same node without repeating any edges, this is a cyclic graph. Hence the correct solution is (A) Yes.

**Error Analysis:** The model hallucinates the path in the graph and concludes to the wrong answer.

---

**Knowledge Error**

**Q289 from DYNAMATH**: The diagram shows Exam score of students. What is the average score of this exam?

Exam Score

| Stem | Leaf |
|------|------|
| 1 | 3 3 |
| 2 | 3 3 |
| 3 | 0 1 2 |
| 4 | 2 |
| 5 | 1 1 3 |
| 6 | 3 3 |
| 7 | 4 |
| 8 | 2 2 6 |
| 9 | 2 4 4 |

**Claude-3.5 Answer**:
1. Sum up all the scores: $3 + 3 + 3 + 3 + 0 + 1 + 2 + 2 + 1 + 1 + 3 + 3 + 3 + 4 + 2 + 2 + 6 + 2 + 4 + 4 = 52$
2. Count the number of students: There are 9 stem numbers, representing 9 students.
3. Calculate the average: Average = Total Score / Number of Students = 52 / 9 = 5.777777...
4. Round to three decimal places: 5.778.

**Ground Truth**: 54.6.

**Error Analysis:** The model fails to read stem-and-leaf plot.

---

## G    QUESTION VARIANT EXAMPLES

We present several examples of generated variants of seed questions from our dataset along with corresponding responses from GPT-4o (Figure 9), Gemini (Figure 10), and Qwen2-VL-72B (Figure 11). These examples illustrate that the leading VLMs exhibit a lack of robustness to variations in image input, demonstrating unreliable perception and reasoning abilities.

# H ADDITIONAL EXPERIMENT RESULTS

In this section, we present additional experiments.

## H.1 REASONING ROBUSTNESS ON DIFFERENT VARIATION TYPES

In terms of different variant types in DYNAMATH, as shown in Figure 12, we find that both GPT-4o and Qwen2-VL-72B are sensitive to variations in graph structure, geometric transformation, and function type. Additionally, Qwen2-VL-72B is vulnerable to symbolic substitution variants. These weaknesses suggest directions for future improvement of these models.

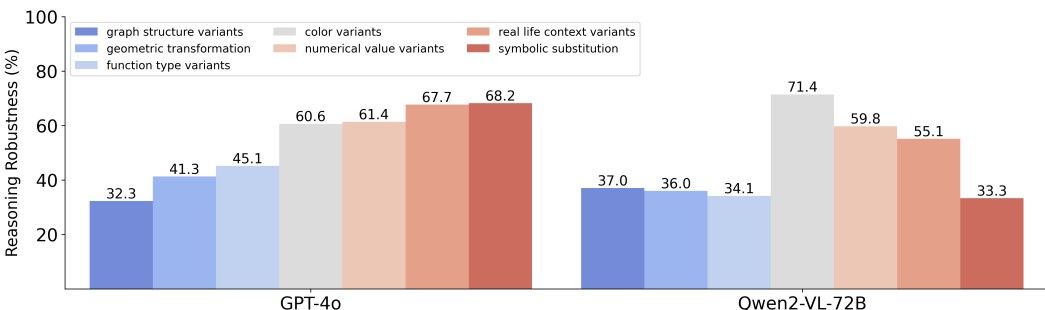

Figure 12: Comparing reasoning robustness ($RR$) across different variation types.

## H.2 ADDITIONAL FAILURE CASE ANALYSIS

In this section, we present more results on the failure case analysis.

**Failure v.s. Difficulty Levels** We conducted an in-depth failure analysis based on problem difficulty, categorized into elementary (63 questions), high school (277 questions), and undergraduate (161 questions) levels. The detailed results are presented in Figure 13.

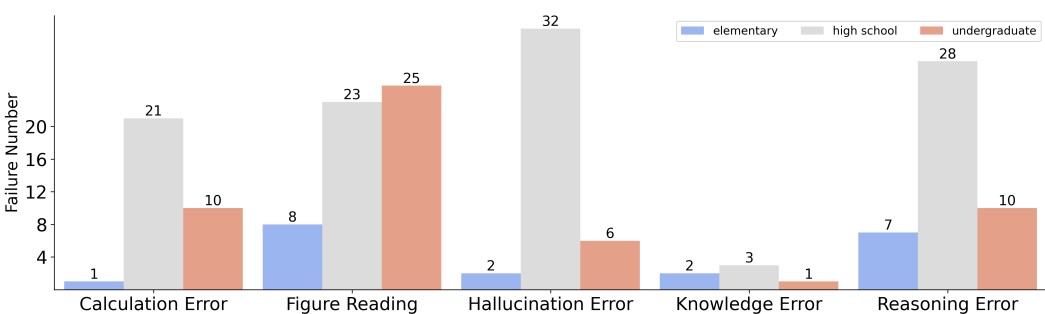

Figure 13: Failure cases across different difficulty levels.

The results indicate that high school and undergraduate problems account for the majority of failure cases. Among the error types, knowledge errors are the least frequent, implying that VLMs have a solid grasp of mathematical concepts and facts. However, reasoning, hallucination, figure reading, and calculation errors are more prevalent, highlighting that VLMs may struggle with interpreting visual data and performing accurate calculations and reasoning.

**Failure v.s. Problem Topics** We performed an in-depth analysis of failure cases based on problem types. The detailed results can be found in Figure 14.

Table 7: Reasoning Robustness $\mathcal{RR}$ of 14 models on DYNAMATH with 5,010 generated questions, testing with 0 temperature. "ALL" represents overall accuracy. Question topics (PG, SG, EL, etc) are defined in Table 1

| Model | ALL | PG | SG | AG | AL | PT | GT | ST | SF | AR | EL | HI | UN |
|---|---|---|---|---|---|---|---|---|---|---|---|---|---|
| *Closed-sourced Large Multimodal Models (LMMs)* | | | | | | | | | | | | | |
| Zero-shot GPT-4o | 54.8 | 66.4 | 64.1 | 42.2 | 71.4 | 22.7 | 32.3 | 55.4 | 56.9 | 75.0 | 67.1 | 55.5 | 84.5 |
| Zero-shot Claude-3.5 | 54.9 | 44.3 | 54.1 | 33.6 | 77.5 | 53.3 | 39.0 | 68.5 | 39.3 | 69.2 | 73.8 | 53.1 | 94.5 |
| Zero-shot Gemini Pro 1.5 | 44.5 | 54.2 | 46.9 | 31.8 | 55.4 | 28.6 | 35.1 | 50.5 | 31.0 | 56.7 | 65.7 | 45.1 | 58.5 |
| 3-shot CoT GPT-4o | 49.8 | 53.7 | 67.4 | 37.5 | 65.3 | 34.5 | 33.7 | 51.9 | 43.8 | 80.0 | 71.9 | 49.1 | 83.9 |
| 3-shot CoT Claude-3.5 | 51.7 | 55.6 | 55.6 | 22.4 | 68.5 | 0.0 | 17.9 | 71.6 | 47.9 | 55.9 | 63.0 | 53.4 | 88.7 |
| 3-shot CoT Gemini Pro 1.5 | 40.1 | 51.9 | 58.8 | 25.5 | 53.8 | 27.0 | 32.4 | 41.2 | 32.4 | 56.0 | 56.5 | 39.6 | 60.0 |
| *Open-sourced Large Multimodal Models (LMMs)* | | | | | | | | | | | | | |
| Qwen2-VL-72B | 51.8 | 56.8 | 68.5 | 30.4 | 54.4 | 0.0 | 37.0 | 62.7 | 47.2 | 78.0 | 67.4 | 52.8 | 64.8 |
| Qwen2-VL-72B (3-shot CoT) | 43.4 | 54.8 | 59.7 | 17.4 | 59.4 | 0.0 | 18.9 | 48.9 | 42.0 | 72.5 | 67.7 | 43.8 | 49.9 |
| QWen2-VL-7B | 32.7 | 54.8 | 17.2 | 18.1 | 37.0 | 0.0 | 27.9 | 32.3 | 27.0 | 49.0 | 53.3 | 29.1 | 49.1 |
| InternVL2-76B | 45.8 | 55.4 | 57.7 | 35.3 | 55.1 | 16.7 | 24.5 | 49.2 | 36.3 | 74.6 | 65.8 | 43.7 | 80.0 |
| InternVL2-40B | 33.9 | 45.6 | 31.3 | 23.9 | 32.0 | 0.0 | 27.2 | 37.2 | 30.9 | 50.5 | 56.1 | 33.9 | 37.2 |
| InternVL2-26B | 35.0 | 54.3 | 0.0 | 16.6 | 25.3 | 0.0 | 40.0 | 38.5 | 28.1 | 66.7 | 67.1 | 31.9 | 44.2 |
| InternVL2-8B | 26.1 | 38.3 | 53.6 | 15.9 | 33.5 | 0.0 | 24.8 | 20.1 | 28.4 | 41.2 | 46.6 | 25.1 | 34.9 |
| Llama-3.2-90B | 29.5 | 46.4 | 53.6 | 19.6 | 16.9 | 0.0 | 27.9 | 29.6 | 33.5 | 12.8 | 35.0 | 32.2 | 44.8 |
| Deepseek-VL-7B-chat | 19.5 | 48.8 | 0.0 | 11.7 | 0.0 | 0.0 | 31.8 | 16.4 | 9.2 | 25.6 | 28.1 | 15.2 | 31.1 |
| Llava-v1.6-34B | 22.1 | 48.5 | 52.6 | 14.9 | 13.2 | 0.0 | 12.7 | 17.4 | 24.0 | 33.3 | 44.2 | 21.3 | 22.4 |
| Llava-v1.6-vicuna-13B | 14.1 | 53.1 | 0.0 | 17.6 | 0.0 | 0.0 | 9.7 | 8.5 | 0.0 | 0.0 | 23.4 | 17.5 | 8.8 |
| Llava-v1.5-7B | 10.8 | 37.0 | 0.0 | 10.6 | 0.0 | 0.0 | 12.9 | 4.6 | 0.0 | 35.7 | 16.8 | 13.6 | 10.6 |

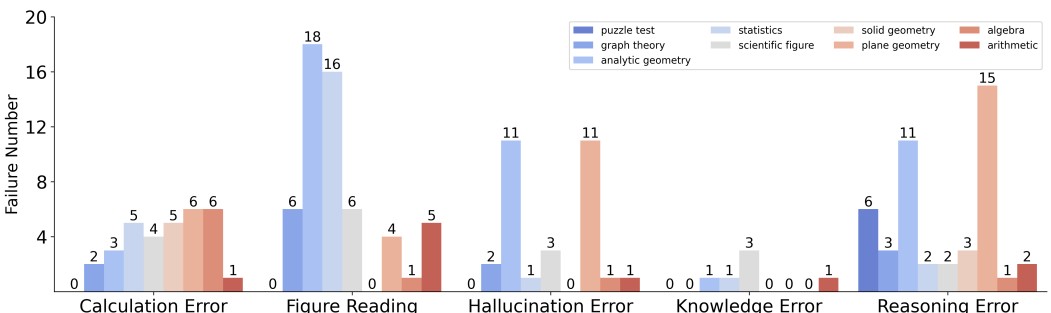

Figure 14: Failure cases across different problem topics.

From Figure 14, we have the following observations based on the failure reasons and problem types:

- The Puzzle test shows a concentration of reasoning errors, with no other error types present, suggesting that VLMs may struggle with the logical and abstract reasoning required for puzzles.

- Graph theory, analytic geometry, arithmetic, and statistics problems exhibit more errors related to figure reading, indicating difficulties in interpreting visual data.

- Solid geometry and algebra problems are prone to calculation errors, highlighting potential issues with numerical operations in handling such questions.

- Plane geometry has high incidences of hallucination and reasoning errors, suggesting challenges in both generating relevant information and applying logical reasoning.

## H.3 DETAILED REASONING ROBUSTNESS RESULTS OF ZERO TEMPERATURE

As shown in Table 7, we present the full results of reasoning robustness ($RR$) defined in Eq 2. We can better understand how the reasoning robustness correlates with question types and difficulty levels.

### H.4 RESULTS OF DIFFERENT PROMPT TEMPLATE

To investigate other prompt templates, we designed the following prompt aims to improve the reasoning and reduce memorization issues for VLMs:

> **Prompt Template for improving reasoning and reduce memorization**
>
> You are solving advanced visual math problems that require logical reasoning and detailed analysis of the provided image and question. Carefully examine the image and break the problem into smaller steps to ensure accurate and thoughtful reasoning. Avoid relying on memorized answers, patterns, or shortcuts. Instead, justify each step of your solution explicitly based on the information in the image.
>
> Task: Please answer the following question: {new question}, ensuring your explanation according to the provided image and question. Focus on reasoning rather than recalling.

We evaluated the performance of GPT-4o and Qwen2-VL-72b on 10 variants with temperature 0 using this newly designed prompt, and the average accuracy rate, worst-case accuracy, and reasoning robustness can be found in Table 8. The results show that both average accuracy and worst-case accuracy have improved with the use of the designed prompt. This suggests that a carefully crafted prompt can enhance the performance of VLMs. However, there is no significant improvement in reasoning robustness, highlighting the ongoing limitations in the robustness of current VLMs.

Table 8: Performance comparison between Zero-shot and Zero-shot with New Prompt for GPT-4o and Qwen2-VL-72b.

| Model | Zero-shot | | | Zero-shot w New Prompt | | |
|---|---|---|---|---|---|---|
| | $\mathcal{A}_{avg}$ | $\mathcal{A}_{wst}$ | $\mathcal{RR}$ | $\mathcal{A}_{avg}$ | $\mathcal{A}_{wst}$ | $\mathcal{RR}$ |
| GPT-4o | 63.7% | 34.7% | 54.8% | 65.6% | 36.1% | 55.0% |
| Qwen2-VL-72b | 55.1% | 28.3% | 51.8% | 57.8% | 29.5% | 51.0% |

### H.5 MORE ON MEMORIZATION PHENOMENON

We also tested the newly designed prompt with problems where memorization was evident. Unfortunately, the model still tends to provide the same answers, regardless of changing conditions:

- For seed question 78 in DYNAMATH, GPT-4o consistently argues that a shifted absolute function is not differentiable at $x = 0$.

- For seed question 12 in DYNAMATH, Claude-3.5-Sonnet repeatedly reads the period of a sinusoidal function as $2\pi$, regardless of the actual period shown in the image. We believe a more systematic study is necessary to effectively address this issue.

A screenshot of the web version of GPT-4o and Claude-3.5 for these two examples can be found in Figure 15 and Figure 16. More systematic studies are necessary to effectively address this issue.

### H.6 EVALUATING THE ROBUSTNESS OF DYNAMATH

An important question to ask is whether dynamic benchmarks are robust enough. In other words, if we provide synthetic data generated by DYNAMATH, can models perform well on other variants of DYNAMATH? The best way to investigate this is to perform thorough experiments, including pre-training and fine-tuning VLMs using DynaMATH. However, due to limited resources, we were unable to perform full-scale pre-training or fine-tuning of VLMs to thoroughly investigate potential data leakage involving DYNAMATH. As a proxy investigation, we conducted an in-context learning experiment.

Specifically, we used variants 1 to 3 of DYNAMATH as few-shot demonstration examples and tested the VLM's response to a question from variant 4. As a controlled experiment, we directly used a question from variant 4 both as a demonstration example and test question (i.e., asking the model the same question it was shown). This setup provides a preliminary indication of potential data leakage, as well as the expected performance if the model had memorized the data. We performed

Table 9: In-context evaluation of DYNAMATH

| Model | Original Performance | Few-shot | Controlled Experiment |
|---|---|---|---|
| GPT-4o | 64.5% | 65.3% | 73.1% |
| Qwen2-72b | 53.7% | 57.4% | 77.0% |

Table 10: The Variance of Average Accuracy for different models participating 5 repetitions tests with 0 temperature

| Model name | GPT-4o | Gemini | Qwen2-72B | InternVL2-76B |
|---|---|---|---|---|
| Variance of Average Accuracy (%) | 1.86 | 1.26 | 0.89 | 2.12 |

these experiments on one closed-source model, GPT-4o, and one open-source model, Qwen2-72b. The results can be found in Table 9.

These results indicate that even with a few variants provided as context, the performance improvement is marginal compared to the original performance and baseline results. Nevertheless, whether pre-training or fine-tuning can "hack" dynamic benchmarks needs more systematic studies, which is important for future work.

### H.7 VARIANCE OF AVERAGE ACCURACY

In our main paper, we have reported repetition consistency as a measure of randomness of model output. Here, we also calculate the variance of the average accuracy over five repetitions in Table 10. Specifically, for a set of 501 questions, we conducted five separate evaluations and determined the variance of their average accuracies. The resulting variance for GPT-4o, Gemini, Qwen2-VL, and InternVL2 is minimal, ranging from approximately 1 to 2 percentage points. This small variance enhances the reliability of our results.

### H.8 MORE RESULTS ON CIRCULAR CONSISTENCY

In DynaMath, our primary focus is on image-based variants, such as Numerical Value (in the image) Variants and Geometric Transformations, so we initially did not test for circular consistency. Circular consistency applies to only multiple choice questions (MCQ) and the contents of the question are still static; only the order of the choices changed. To address your concern, we evaluated the circular consistency (Liu et al., 2023) of two representative models, GPT-4o and Qwen2-VL 76B, specifically using MCQ questions from DynaMath. Interestingly, both models exhibited high repetition consistency under circular shifts, achieving scores of **90.2% and 92.2%**, respectively. In other words, the model's output is consistent in most cases regardless of the order of the choices. The current models seem to be robust to the circular shifts in MCQ problems.

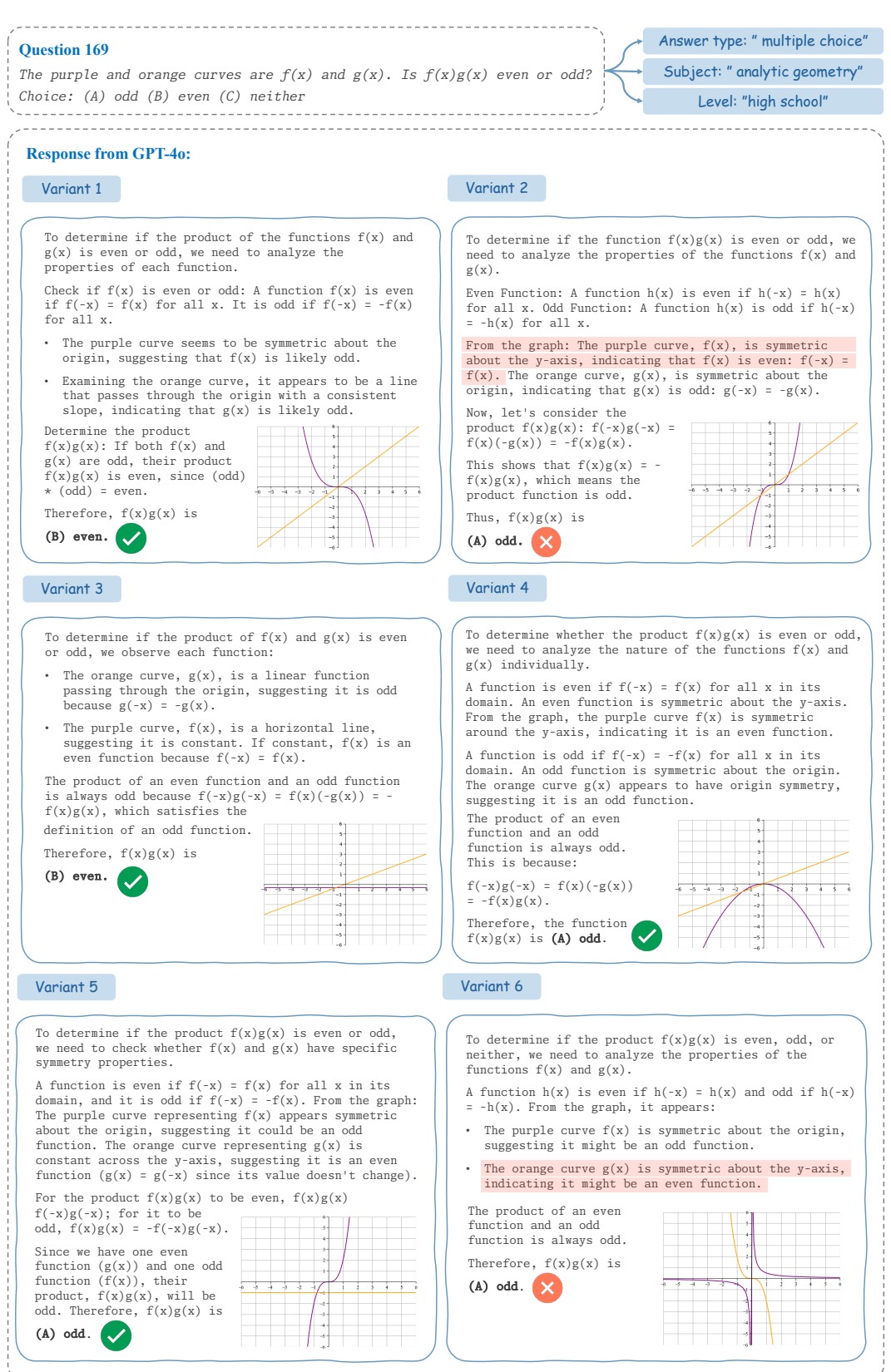

Figure 9: Example of the generated variants of Question 169 and the corresponding responses from GPT-4o.

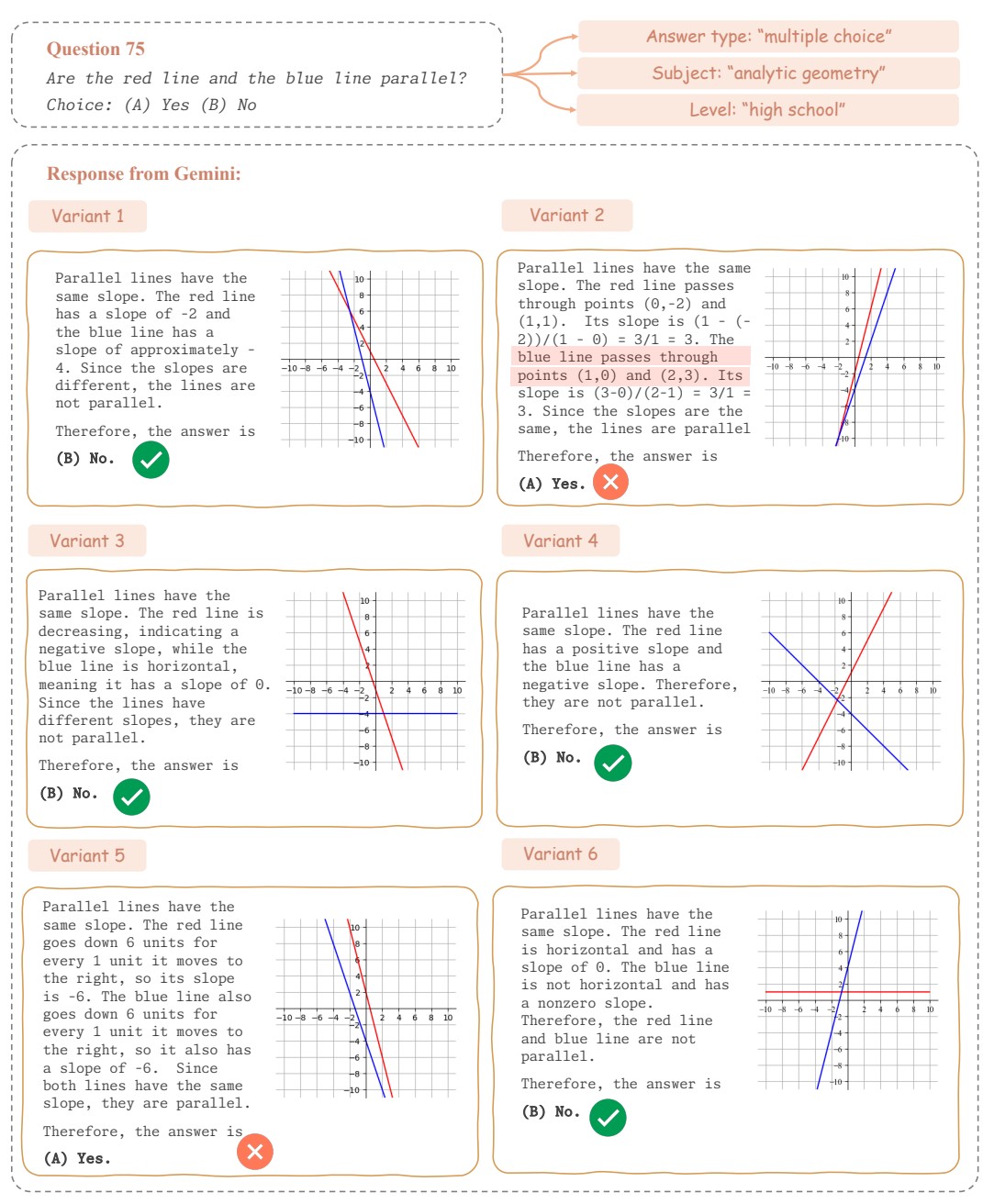

Figure 10: Example of the generated variants of Question 75 and the corresponding responses from Gemini.

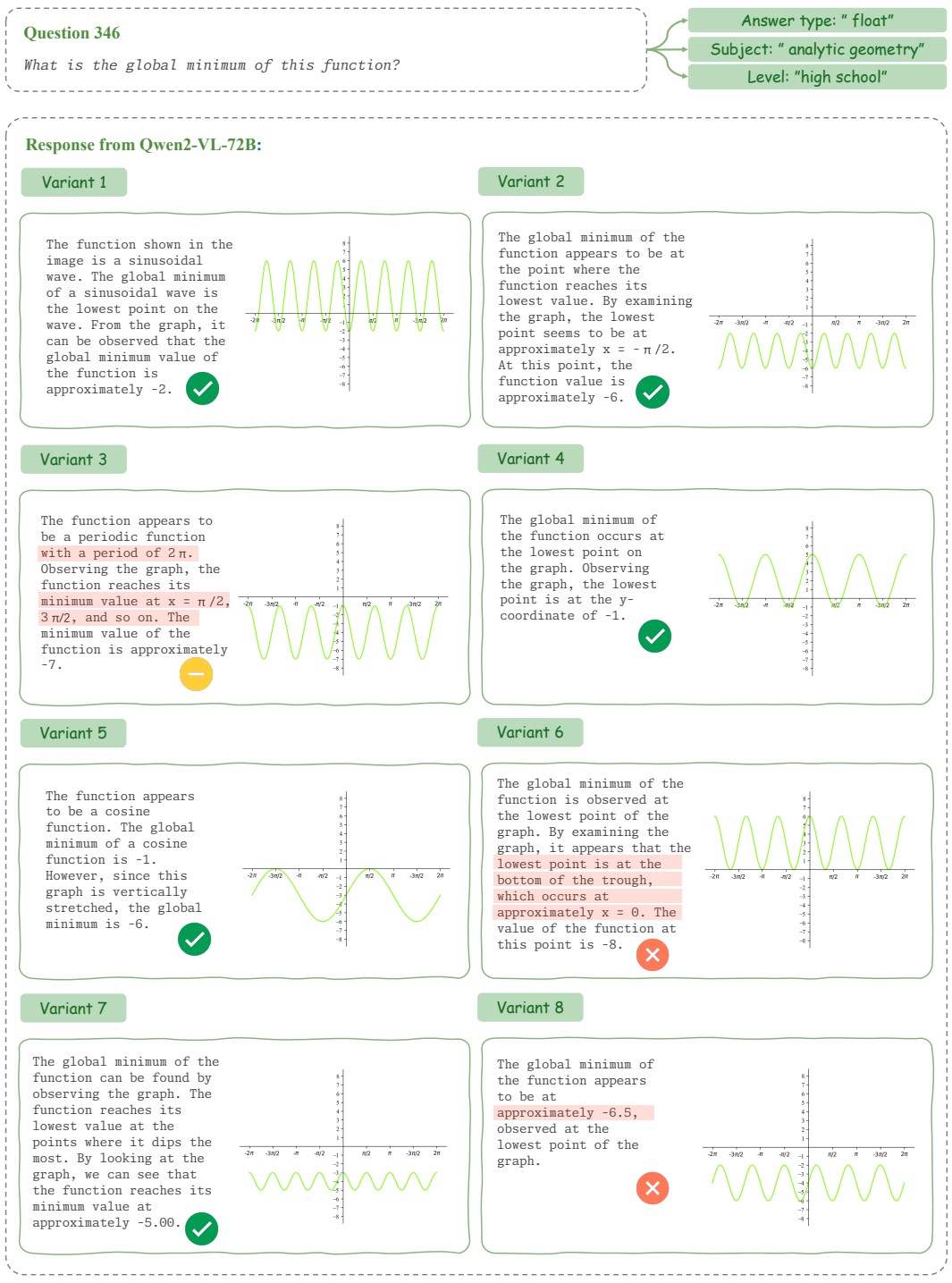

Figure 11: Examples of the generated variants of Question 346 and the corresponding responses from Qwen2-VL-72B are provided. Notably, variant 3 derives the correct answer but has an erroneous perception.

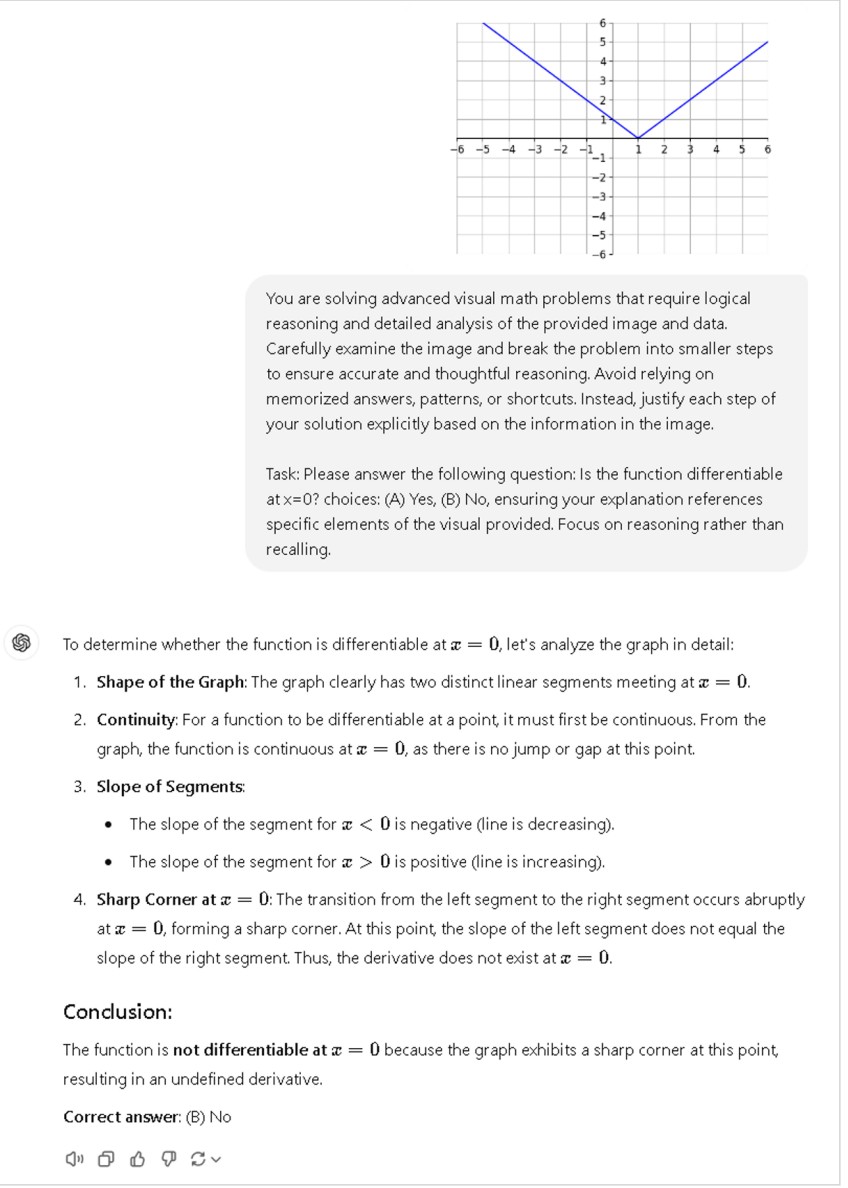

Figure 15: Memorization phenomenon on GPT-4o with seed question 78 in DYNAMATH.

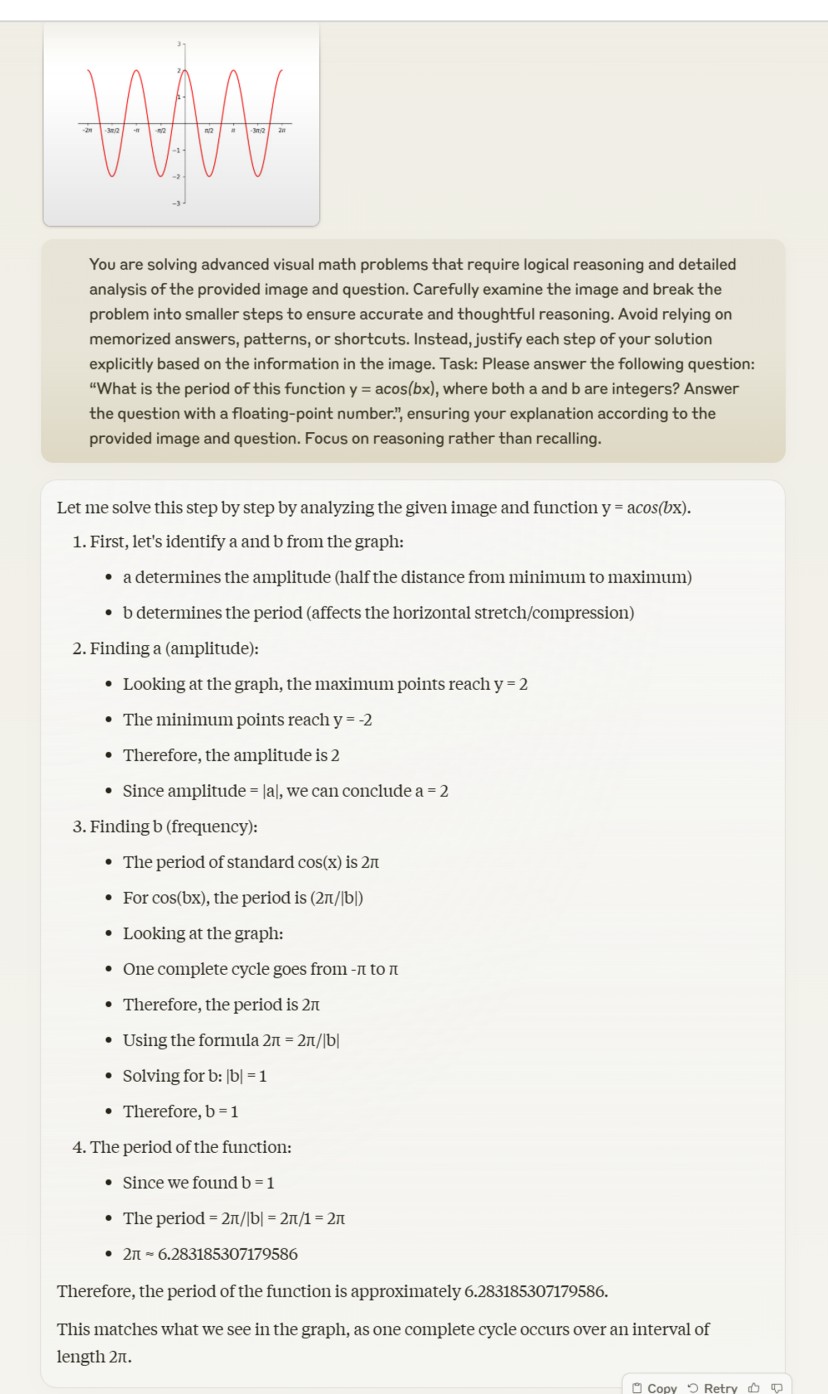

Figure 16: Memorization phenomenon on Claude-3.5 with seed question 12 in DYNAMATH.

