# OpenReview forum: "DynaMath: A Dynamic Visual Benchmark for Evaluating Mathematical Reasoning Robustness of Vision Language Models"
_ICLR.cc/2025/Conference — ICLR 2025 Poster_

### Official Review · Reviewer_Nqb6 · 2024-10-27

**Soundness:** 3
**Presentation:** 3
**Contribution:** 3
**Rating:** 8
**Confidence:** 4

**Summary:**

The paper introduces DYNAMATH, a dynamic visual math benchmark aimed at evaluating the mathematical reasoning robustness of Vision-Language Models (VLMs).
The benchmark reveals that state-of-the-art VLMs, like GPT-4o, struggle with mathematical reasoning when visual numerical values or function graphs change, despite these tasks being simple for humans.
DYNAMATH includes 501 seed questions, each represented as a Python program, capable of generating varied concrete questions with different visual and textual alterations.
The benchmark was used to assess 14 VLMs (including proprietary ones and open-source ones) with 5,010 generated questions, showing a significant gap between average-case and worst-case model accuracy.
The paper's contributions include identifying the weakness in VLMs' reasoning abilities, introducing DYNAMATH for robustness evaluation, and providing insights to guide the development of more reliable VLMs for mathematical reasoning.

**Strengths:**

1. DYNAMATH is the first dynamic visual math benchmark designed to evaluate the robustness of mathematical reasoning in VLMs.
The dynamic generation of varied problem sets more accurately simulates real-world problem-solving, providing a more genuine assessment of model capabilities.
2. With 501 high-quality seed questions that can generate a multitude of variant problems across various mathematical topics and difficulty levels, DYNAMATH allows for an in-depth evaluation of VLMs' generalization abilities under different input conditions, revealing the limitations of current models when dealing with problem variations.

**Weaknesses:**

1. Lacks analysis of open-source VLMs performance under the zero-shot / few-shot setting.
2. Lacks analysis of the reproducibility of evaluation results (since temperature is not 0).

**Questions:**

1. What about the zero-shot / few-shot COT results for open-source VLMs? It would be beneficial to present those results and do some analysis.
2. Evaluating VLMs with temperature not set to 0 will make the results less reproducible. Will the evaluation result vary much when you run the evaluation multiple times? Can you provide an error bar for it?
3. Would you please provide the detailed distribution of problems in DYNAMATH (multiple choice, answer is float, answer is text)?
4. In MMBench [1], the authors have defined another consistency named circular consistency? Will the Repetition Consistency still be that high (>80%) when circular shift is applied to choices for MCQ problems in DYNAMATH?

[1] MMBench: Is Your Multi-modal Model an All-around Player?

---

> ### Author Response · Authors · 2024-11-14
> **Missing Weaknesses Section**
>
> Dear Reviewer Nqb6,
>
> We appreciate the time and effort you took to review our submission and thank you for your insightful questions. Upon reading your comments, we noticed that the **Weaknesses** section is missing probably due to a technical issue of OpenReview. Could you please clarify if there were any missing points that you would like to bring to our attention? If so, we would greatly appreciate it if you could provide additional feedback on these aspects and we would like to address your concerns fully during the rebuttal. Thank you.
>
> Sincerely,
>
> Authors

---

> > ### Comment · Reviewer_Nqb6 · 2024-11-14
> >
> > Dear Authors,
> > I have updated the weakness section.

---

> > > ### Author Response · Authors · 2024-11-14
> > >
> > > Dear Reviewer Nqb6, thank you for your prompt clarification. We will address your concerns and questions in our following posts.
> > >
> > > Sincerely,
> > >
> > > Authors

---

> ### Author Response · Authors · 2024-11-22
> **Author Response (part 1/2)**
>
> We appreciate the reviewers' valuable comments and have provided our responses below. Based on your suggestions, we have reran all experiments with zero temperature and **added new experiments** on COT of open-source VLMs and circular consistency evaluation. We hope these responses address your concerns, and we welcome any further feedback.
>
> **Q1. What about the zero-shot / few-shot COT results for open-source VLMs? It would be beneficial to present those results and do some analysis.**
>
> **A1:**  We now included the results of **3-shot COT for Qwen2-VL-76B in Table 1 and Table 2**. The results continue to confirm that there is no consistent improvement across different VLMs, which showed that Qwen2-VL-76B experienced a decline in performance with COT. This finding aligns with previous evaluation research for VLMs [1].
>
> | Model | ALL | PG | SG | AG | AL | PT | GT | ST | SF | AR |
> |------------------------------|------|-----|-----|-----|-----|-----|-----|-----|-----|-----|
> | Zero-shot Qwen2-VL-72B (Average-case Accuracy) | 55.1 | 48.1| 48.7| 50.9| 57.6| 28.2| 45.0| 68.9| 56.4| 54.2|
> | 3-shot COT Qwen2-VL-72B (Average-case Accuracy) | 52.4 | 45.1| 44.7| 47.5| 59.4| 19.4| 44.2| 67.1| 52.9| 53.1|
> | Qwen2-VL-72B (Worst-case Accuracy) | 28.3 | 27.3| 33.3| 15.5| 31.4| 0.0 | 16.7| 43.2| 26.7| 42.3|
> | 3-shot COT Qwen2-VL-72B (Worst-case Accuracy) | 22.8 | 24.7| 26.7| 8.2 | 35.3| 0.0 | 8.3 | 32.8| 22.2| 38.5|
>
>
>
> **Q2. Evaluating VLMs with temperature not set to 0 will make the results less reproducible. Will the evaluation result vary much when you run the evaluation multiple times? Can you provide an error bar for it?**
>
> **A2:** Thanks for the question. We also noticed the potential impact of the temperature setting on our results. In response to your feedback, we have **re-conducted all experiments with the temperature set to $\tau=0$ for all models, which demonstrates the same trend** but a slight improvement on the scores due to reduced randomness. The reduced temperature also leads to higher repetition consistency, the metric we utilized to measure the inherent randomness in model outputs. This further strengthens our observation that models can be consistently wrong on certain variants, such as those we demonstrated in Figure 1. As shown in Table 4, the repetition consistency for GPT-4o and Qwen2-VL-72B increases from 86.79% and 84.23% to **94.1% and 98.9%**, respectively. This adjustment also makes our results more reproducible and confirms that the evaluation is consistent across multiple runs with high probability.
>
> We also address your concern regarding error bars by calculating the **variance of the average accuracy** over five repetitions in the below table. Specifically, for a set of 501 questions, we conducted five separate evaluations and determined the variance of their average accuracies. The resulting variance for GPT-4o, Gemini, Qwen2-VL, and InternVL2 is minimal, ranging from approximately 1 to 2 percentage points. This small variance enhances the reliability of our results.
>
>
> | Metric | GPT-4o | Gemini | Qwen2-72B | InternVL2-76B |
> |--------------------------------------|--------|--------|-----------|---------------|
> | Variance of Average Accuracy (%) | 1.86 | 1.26 | 0.89 | 2.12 |
>
>
>
> **Q3. Would you please provide the detailed distribution of problems in DYNAMATH (multiple choice, answer is float, answer is text)?**
>
> **A3:** In Table 1, we have categorized the DynaMath question types into multiple-choice and free-form text (including float and integer formats). To clearly differentiate numerical answers from free-form text answers, we present the distribution of problems below, with updates reflected in Table 1.
>
>
> | Problem Type        | Percentage |
> |---------------------|------------|
> | Numerical           | 59.1%     |
> | Multiple Choice     | 34.7%     |
> | Free-form Text      | 6.2%      |

---

> ### Author Response · Authors · 2024-11-22
> **Author Response (part 2/2)**
>
> **Q4. In MMBench, the authors have defined another consistency named circular consistency? Will the Repetition Consistency still be that high (>80%) when circular shift is applied to choices for MCQ problems in DYNAMATH?**
>
> **A4:** Thank you for bringing this related method evaluating VLM robustness to our attention. We added reference and discussed the difference in related work (Section 2). In DynaMath, our primary focus is on image-based variants, such as Numerical Value (in the image) Variants and Geometric Transformations, so we initially did not test for circular consistency. Circular consistency applies to only multiple choice questions (MCQ) and the contents of the question are still static; only the order of the choices changed.
>
> To address your concern, we evaluated the circular consistency [2] of two representative models, GPT-4o and Qwen2-VL 76B, specifically using MCQ questions from DynaMath. Interestingly, both models exhibited high repetition consistency under circular shifts, achieving scores of **90.2% and 92.2%**, respectively. In other words, the model’s output is consistent in most cases regardless of the order of the choices. The current models seem to be robust to the circular shifts in MCQ problems.
>
> **References**
>
> [1] Wang K, Pan J, Shi W, et al. Measuring multimodal mathematical reasoning with math-vision dataset[J]. arXiv preprint arXiv:2402.14804, 2024.
>
> [2] MMBench: Is Your Multi-modal Model an All-around Player?

---

> ### Comment · Reviewer_Nqb6 · 2024-11-26
>
> The rebuttal well resolved my previous concerns. Thus I would like to raise my final rating.

---

> > ### Author Response · Authors · 2024-11-26
> > **Thank You for Your Feedback**
> >
> > We sincerely thank the reviewer for their thoughtful and encouraging feedback. We are delighted that our responses have successfully addressed the concerns raised and appreciate the reviewer’s support for our work.

---

### Official Review · Reviewer_Fgu5 · 2024-10-30

**Soundness:** 4
**Presentation:** 4
**Contribution:** 4
**Rating:** 8
**Confidence:** 5

**Summary:**

DynaMATH, a new benchmark for evaluating visual reasoning in vision language models, offers greater controllability than previous tools like MathVistas. Using 501 programmable seed questions with multiple variants, it tests both closed and open-source VLMs, revealing insights about worst-case accuracy and memorization patterns.

**Strengths:**

1. The paper effectively presents comprehensive and interesting findings.
2. The benchmark's controllability should be highlighted as a key strength.

**Weaknesses:**

1. Move Figure 7 to the main paper to better illustrate the benchmark's appearance.
2. The paper lacks details about the seed question curation process and should address the benchmark's scalability limitations.

**Questions:**

Can the authors provide a rough time estimate for designing/redesigning seed questions? This is important for future benchmark design. Also, how can we scale up the design flow for a larger dynamic math benchmark?

---

> ### Author Response · Authors · 2024-11-22
> **Author Response**
>
> We appreciate the reviewers' valuable comments and have provided our responses below. We hope these responses address your concerns, and we welcome any further feedback.
>
> **Q1: Move Figure 7 to the main paper to better illustrate the benchmark's appearance.**
>
> A1: We appreciate the reviewer's suggestion, and in response, we have relocated a portion of this figure to the main paper (Figure 3) for easier reading.
>
> **Q2: How can we scale up the design flow for a larger dynamic math benchmark?**
>
> A2: We thank the reviewer for this insightful comment. Please refer to General Response 2 on the scalability of DynaMATH, where we highlight that one key challenge in scaling up DynaMATH is the inclusion of dynamic visual elements. One needs to carefully design the drawing program so that the generated images meet the expectations. In addition, we propose a potential solution to scale up through LLM-assisted generation. LLMs have shown proficiency in generating text-based problems and writing code. It is possible to break down benchmark topics and subtopics, prompting the LLM to generate diverse problem sets and corresponding Python programs for visual elements. The generated problems should be dynamic, with parameterizable Python code to produce multiple image variants. To this end, DynaMATH is a valuable benchmark since our seed questions can serve as high-quality human demonstrations to guide the LLMs for this task. This LLM-assisted approach could significantly reduce manual effort. However, some human intervention will still be necessary to ensure the selection of correct and high-quality samples from LLMs.
>
> While we have to leave the LLM-assisted dynamic benchmark generation as a future work, DynaMATH can serve as a good baseline which is completely crafted by human beings, and future work on automated dynamic benchmark generation may compare to DynaMATH in terms of diversity and quality.
>
> **Q3: Can the authors provide a rough time estimate for designing/redesigning seed questions? This is important for future benchmark design.**
>
> A3: Designing a seed question typically takes 30 to 60 minutes, depending on the complexity of the problem and the degree of randomization required. For example, simple arithmetic or algebraic questions can be designed quickly, while problems involving intricate logic or geometric variants require more time and careful consideration in the program design.

---

### Official Review · Reviewer_GLTU · 2024-11-05

**Soundness:** 3
**Presentation:** 3
**Contribution:** 2
**Rating:** 6
**Confidence:** 5

**Summary:**

This paper introduces a dynamic visual benchmark designed to evaluate the mathematical reasoning robustness of VLMs. Unlike static benchmarks, DynaMath generates question variations programmatically to test models under diverse conditions. This allows for a nuanced assessment of VLMs’ robustness and generalization across mathematical tasks.

**Strengths:**

- The proposed dynamic benchmark for evaluating the mathematical reasoning robustness of VLMs are complementary to existing benchmarks, and it aligns well with the challenges in real-world mathematical applications, where variations in problem conditions often arise.
- Extensive evaluations on reasoning robustness are conducted and some observations are beneficial to the community.

**Weaknesses:**

- Although the paper presents some error analysis, the model robustness is still underexplored. Quite a few factors are not investigated, such as the difficulty level and type of questions and variants.
- Besides the question itself, the prompt template can also result in different responses. The robustness regarding to prompt variants is neither studied nor excluded from the research scope with clear explanation.
- Deeper insights into the failure modes and interpretability would be great.

**Questions:**

1. It is uncertain if the benchmark could be easily hacked by just adopting the same program-based question generation to produce training data and then finetune the models.
2. Is the reasoning robustness relevant to the difficulty level and type of questions and variants?
3. Is there any preliminary study on how to scale the dynamic generation approach for more comprehensive and difficult benchmarks?
4. It would be insightful to know if the authors have tested any interventions (e.g., prompt adjustments) to increase the robustness and reduce the memorization phenomenon.
5. Is there any interpretability of the observations on the mathematical reasoning robustness?

---

> ### Author Response · Authors · 2024-11-22
> **Author Response (part 1/4)**
>
> We appreciate the reviewers' valuable comments and have provided our responses below. In response to your suggestions, we have conducted **new experiments and analyses**, including: 1. an examination of model reasoning robustness and failure modes across various problem topics, variants, and difficulty levels; 2. an evaluation of performance under different prompt templates; and 3. a preliminary study on the robustness of DynaMATH in the context of data leakage issues. We hope these responses address your concerns, and we welcome any further feedback.
>
> **Q1: Model robustness across problem difficulty levels, type of questions**
>
> A1: We thank the reviewer for this question. In response, we have conducted a new experiment on evaluating the reasoning robustness of 10 variants of DynaMATH and 14 evaluated VLMs according to different problem types and difficulty levels. The detailed results can be found in Table 7 at Appendix H.3 in the revised paper. Due to space constraints, we list the detailed results for GPT-4o and Qwen2-VL-72b as below:
>
> | Model         | Problem Topics                |                    |                    |                    |                    |                    |                    |                    |                    | Difficulty Levels          |                    |                    |                    |
> |---------------|--------------------------------|--------------------|--------------------|--------------------|--------------------|--------------------|--------------------|--------------------|--------------------|----------------------------|--------------------|--------------------|--------------------|
> |               | Plane Geometry (PG)          | Solid Geometry (SG)| Analytic Geometry (AG)| Algebra (AL)      | Puzzle Test (PT)   | Graph Theory (GT)  | Statistics (ST)    | Scientific Figure (SF)| Arithmetic (AR)  | Elementary School (EL)    | High School (HI)  | Undergraduate (UN)|
> | GPT-4o        | 66.4                          | 64.1               | 42.2               | 71.4               | 22.7               | 32.3               | 55.4               | 56.9               | 75.0               | 67.1                      | 55.5               | 84.5               |
> | Qwen2-VL-72b  | 56.8                          | 68.5               | 30.4               | 54.4               | 0.0                | 37.0               | 62.7               | 47.2               | 78.0               | 67.4                      | 52.8               | 64.8               |
>
> From the above results, we make the following observations:
>
> GPT-4o generally outperforms Qwen2-VL-72b across most problem topics and difficulty levels, indicating a stronger reasoning robustness.
>
> Among different problem topics, both models perform well on Arithmetic  Arithmetic (AR), but show lower robustness on Graph Theory (GT) problem types, indicating challenges in reasoning with graph transformations. In addition, both models struggle with Puzzle Test (PT) problems, highlighting a major area for improvement in logical and abstract reasoning.
>
> Among different difficulty levels, both models perform well on Elementary School  (EL) problems, suggesting that they handle easier problems effectively. On the other hand, GPT-4o shows strong performance on Undergraduate  (UN) problems.
>
> In summary, the varied performance across topics and difficulty levels highlights the ongoing challenges in achieving consistent reasoning robustness across different problem types and difficulty levels.
>
> **Q2: Model robustness across different variants.**
>
> A2: We performed the analysis on the relevance of the reasoning robustness with the problem variants on GPT-4o and Qwen2-VL-72b and added this new analysis to Appendix H.1 in the revised paper, the results on reasoning robustness (defined in Eq 2, higher is better) can be found as below.
>
> | Variants Type | Graph Structure | Geometric Transformations | Function Types | Color Variants | Numerical Value Variants | Real-life Content Variants | Symbolic Substitution |
> |---------------|-----------------|---------------------------|----------------|----------------|--------------------------|---------------------------|-----------------------|
> | GPT-4o        | 32.3           | 41.3                      | 45.1           | 60.6           | 61.4                     | 67.7                      | 68.2                  |
> | Qwen2-VL-72b  | 37.0           | 36.0                      | 34.1           | 71.4           | 59.8                     | 55.1                      | 33.3                  |
>
>
>
> From the above results, it can be seen that both open-source model and closed-source model have less robustness when handling graph-based variants. GPT-4o (closed-source) demonstrates higher robustness in handling color variants, numerical value variants, real-life content variants, and symbolic substitutions, while Qwen2-VL-72b (open source model) has less robustness on the symbolic substitutions.

---

> ### Author Response · Authors · 2024-11-22
> **Author Response (part 2/4)**
>
> **Q3: In depth failure analysis across different problem difficulty levels**
>
> A3: We thank the reviewer for the constructive feedback. We have conducted an in-depth failure analysis based on problem difficulty, categorized into elementary (easy), high school (moderate), and undergraduate (difficult) levels and added this new analysis in Appendix H.2 in the revised paper. The detailed results are presented below:
>
> | Difficulty Level       | Calculation Error | Figure Reading | Hallucination Error | Knowledge Error | Reasoning Error |
> |-------------------------|-------------------|----------------|---------------------|-----------------|-----------------|
> | Elementary (63 questions) | 1                 | 8              | 2                   | 2               | 7               |
> | High School (277 questions) | 21                | 23             | 32                  | 3               | 28              |
> | Undergraduate (161 questions) | 10                | 25             | 6                   | 1               | 10              |
>
>
> The results indicate that high school and undergraduate problems account for the majority of failure cases, suggesting that VLMs encounter significant challenges as problem difficulty increases. Among the error types, knowledge errors are the least frequent, implying that VLMs have a solid grasp of mathematical concepts and facts. However, reasoning, hallucination, figure reading, and calculation errors are more prevalent, highlighting that VLMs may struggle with interpreting visual data and performing accurate calculations and reasoning.
>
> **Q4: In depth failure analysis based on problem topics.**
>
> A4: We thank the reviewer for the comment, we have performed an in-depth analysis of failure cases based on problem types and added this new analysis to Appendix H.2 in the revised paper. The detailed results can be found as below:
> | Topics                   | Calculation Error | Figure Reading | Hallucination Error | Knowledge Error | Reasoning Error |
> |--------------------------|-------------------|----------------|---------------------|-----------------|-----------------|
> | Puzzle Test (17 questions)  | 0                 | 0              | 0                   | 0               | 6               |
> | Graph Theory (48 questions) | 2                 | 6              | 2                   | 0               | 3               |
> | Analytic Geometry (97 questions) | 3             | 18             | 11                  | 1               | 11              |
> | Statistics (125 questions)    | 5                 | 16             | 1                   | 1               | 2               |
> | Scientific Figure (45 questions) | 4             | 6              | 3                   | 3               | 2               |
> | Solid Geometry (15 questions) | 5                 | 0              | 0                   | 0               | 3               |
> | Plane Geometry (77 questions) | 6                | 4              | 11                  | 0               | 15              |
> | Algebra (51 questions)     | 6                 | 1              | 1                   | 0               | 1               |
> | Arithmetic (26 questions)  | 1                 | 5              | 1                   | 1               | 2               |
>
>
> From the above results, we have the following observations based on the failure reasons and problem types:
>
> 1.Puzzle test shows a concentration of reasoning errors, with no other error types present, suggesting that VLMs may struggle with the logical and abstract reasoning required for puzzles.
>
> 2.Graph theory, analytic geometry, arithmetic, and statistics problems exhibit more errors related to figure reading, indicating difficulties in interpreting visual data.
>
> 3.Solid geometry and algebra problems are prone to calculation errors, highlighting potential issues with numerical operations on handling such questions.
>
> 4.Plane geometry has high incidences of hallucination and reasoning errors, suggesting challenges in both generating relevant information and applying logical reasoning.

---

> ### Author Response · Authors · 2024-11-22
> **Author Response (part 3/4)**
>
> **Q5: Model performance with different prompt templates to improve the reasoning ability.**
>
> A5: We appreciate the reviewer’s insightful comment. In our original paper, we adopted the standard chain-of-thought (CoT) prompt techniques. To investigate other prompt template, we designed the following prompt aims to improve the reasoning and reduce memorization issues for VLMs:
> ```
> You are solving advanced visual math problems that require logical reasoning and detailed analysis of the provided image and question. Carefully examine the image and break the problem into smaller steps to ensure accurate and thoughtful reasoning. Avoid relying on memorized answers, patterns, or shortcuts. Instead, justify each step of your solution explicitly based on the information in the image.
>
> Task: Please answer the following question: {new_question}, ensuring your explanation according to the provided image and question. Focus on reasoning rather than recalling.
>
> ```
> We evaluated the performance of GPT-4o and Qwen2-VL-72b on 10 variants with temperature 0 using this newly designed prompt, and the average accuracy rate (defined in Eq 1), worst-case accuracy (defined in Eq 1), and reasoning robustness (defined in Eq 2) can be found in the following table.
>
> | Model         | Zero-shot            |                        |                        | Zero-shot w New Prompt  |                        |                        |
> |---------------|----------------------|------------------------|------------------------|-------------------------|------------------------|------------------------|
> |               | $\mathcal{A}_{avg}$ (↑) | $\mathcal{A}_{wst}$ (↑)   | $\mathcal{RR}$ (↑)     | $\mathcal{A}_{avg}$ (↑) | $\mathcal{A}_{wst}$ (↑) | $\mathcal{RR}$ (↑)     |
> | GPT-4o        | 63.7%                | 34.7%                  | 54.8                  | 65.6%                   | 36.1%                  | 55.0                  |
> | Qwen2-VL-72b  | 55.1%                | 28.3%                  | 51.8                  | 57.8%                   | 29.5%                  | 51.0                  |
>
> The results show that both average accuracy and worst-case accuracy have slightly improved with the use of the designed prompt. This suggests that a carefully crafted prompt can enhance the performance of VLMs. However, there is no significant improvement in reasoning robustness, highlighting the ongoing limitations in the robustness of current VLMs.
>
>
> **Q6: Prompt to reduce the memorization.**
>
> A6: We thank the reviewer for raising this question. We also tested the new designed prompt (mentioned in Q5 above) with problems where memorization was evident. Unfortunately, the model still tends to provide the same answers, regardless of changing conditions:
>
> - For seed question 78 in DynaMATH, GPT-4o consistently argues that a shifted absolute function is not differentiable at $x=0$.
>
> - For seed question 12 in DynaMATH, Claude-3.5-Sonnet repeatedly reads the period of a sinusoidal function as pi or $2\pi$, regardless of the actual period shown in the image.
>
> A screenshot of the web-version of GPT-4o and Claude-3.5 for these two examples can be found in Figure 15 in the revised paper. We believe a more systematic study is necessary to effectively address this issue.

---

> > ### Author Response · Authors · 2024-11-22
> > **Author Response (part 4/4)**
> >
> > **Q7: It is uncertain if the benchmark could be easily hacked by just adopting the same program-based question generation to produce training data and then fine-tune the models.**
> >
> > A7: We thank the reviewer for the constructive feedback. Due to limited resources, we were unable to perform full-scale pre-training or fine-tuning of Vision-Language Models (VLMs) to thoroughly investigate potential data leakage involving DynaMATH. However, we **conducted an in-context learning experiment** as a proxy investigation.
> >
> > In this experiment, we used variants 1 to 3 from the 10 generated variants of DynaMATH as few-shot demonstration examples and tested the VLM’s response on a question from variant 4. As a controlled experiment, we directly used a question from variant 4 both as a demonstration example and test question (i.e., asking the model the same question it was shown). This setup provides a preliminary indication of potential data leakage, as well as the expected performance if the model had memorized the data. We performed these experiments on one close-sourced model GPT-4o and one open-source model Qwen2-72b.
> >
> > | Model         | Original Performance | Few-shot (query a new variant) | Controlled Experiment |
> > |---------------|----------------------|---------------------------------|--------------------------------------|
> > | GPT-4o        | 64.5%               | 65.3%                          | 73.1%                               |
> > | Qwen2-72b     | 53.7%               | 57.4%                          | 77.0%                               |
> >
> >
> > These results indicate that even with a few variants provided as context, the performance improvement is marginal compared to the original performance and controlled experiment results. Nevertheless, whether pre-training or fine-tuning can “hack” dynamic benchmarks needs more systematic studies, which is an important future work.
> >
> > **Q8: Is there any preliminary study on how to scale the dynamic generation approach for more comprehensive and difficult benchmarks?**
> >
> > A8: We thank the reviewer for this insightful comment. We agree that studying how to scale the dynamic generation process for more comprehensive and challenging benchmarks is crucial. Please refer to General Response 2 on the scalability of DynaMATH, where we highlight the key challenges in scaling up and propose potential solutions through LLM-assisted generation. A more systematic study is needed to thoroughly investigate the scalability of dynamic benchmark development.

---

> > > ### Comment · Reviewer_GLTU · 2024-11-25
> > >
> > > I appreciate the author's feedback and it addressed my concerns. I will maintain my rating.

---

> > > > ### Author Response · Authors · 2024-11-26
> > > > **Thank You for Your Feedback**
> > > >
> > > > Thank you for taking the time to review our work and for providing feedback. We are delighted that our responses have effectively addressed your concerns and sincerely appreciate your continued support. If you have any additional comments or suggestions, please feel free to share them, we would be happy to discuss them further.

---

### Official Review · Reviewer_bDGf · 2024-11-09

**Soundness:** 3
**Presentation:** 3
**Contribution:** 3
**Rating:** 6
**Confidence:** 5

**Summary:**

This paper introduces DYNAMATH, a dynamic visual benchmark designed to evaluate the mathematical reasoning robustness of Vision-Language Models (VLMs). DYNAMATH contains 501 seed questions implemented as Python programs that can generate diverse variants by altering visual conditions while maintaining the same underlying mathematical concepts. The benchmark spans multiple mathematical topics and difficulty levels. The authors evaluated 14 state-of-the-art VLMs (both closed and open-source) and found significant gaps between average-case and worst-case performance, indicating limitations in VLMs' mathematical reasoning robustness.

**Strengths:**

1. The dynamic nature of the benchmark is innovative and addresses a critical gap in existing evaluation methods for VLMs' mathematical reasoning capabilities.
2. The evaluation is comprehensive. The author tests multiple aspects of mathematical reasoning across 9 different topics and they evaluate both closed and open-source models. The detailed performance includes various types of variations (numerical, geometric, functional, etc.).
3. The ablation analysis is strong: (a) detailed performance breakdown by topic and difficulty level;(b) Introduction of meaningful metrics (worst-case accuracy, reasoning robustness);(3)Thorough error analysis with clear categorization
4. The methodology is well-documented. The author provides a clear explanation of benchmark creation process.

**Weaknesses:**

1. The authors acknowledge that the difficulty level is relatively limited compared to some existing benchmarks like MATH-V. The requirement for programmatic generation may restrict the inclusion of more complex problems
2. The selection of seed questions and their variants might introduce unintended biases. The paper doesn't deeply discuss potential limitations in the variation generation process.
3. The human evaluation seems limited (only 10 participants). It is unclear if the human evaluators' backgrounds are representative, which may limit the validation of human performance.
4. Generalization of the conclusion on realistic questions. The questions (especially the vision content) are created by the program, which is quite different from the figures in the application (like the exam paper; hand-written question, or photo of the book). If the model fits well on these tasks, does it mean the model performs well on the realistic scenario?
5.Missing in-depth analysis of the performance gap between the worst-case and the standard accuracy. If we use a large amount of synthetic data in a similar manner, can the model work well also on these problems? If the performance gap can be resolved by introducing more synthetic data, the conclusion will be trivial and have no interesting news.

**Questions:**

In addition to the problems in weaknesses. There are some extra questions and suggestions.
1. Can we convert the existing problem into a dynamic/programmatic version?
2. Including analysis of the correlation between different types of variations and model performance will be better.
3. Investigate potential relationships between training data/training techniques and performance on different variation types if possible.

---

> ### Author Response · Authors · 2024-11-22
> **Author Response (part 1/2)**
>
> We appreciate the reviewers' valuable comments and have provided our responses below. Based on your suggestions, we have **added new analysis** on the correlation between different variant types and reasoning robustness, as well as additional human evaluations. We hope these responses address your concerns, and we hope you can reevaluate our paper based on our new results and we also welcome any further feedback.
>
> **Q1. The authors acknowledge that the difficulty level is relatively limited compared to some existing benchmarks like MATH-V. The requirement for programmatic generation may restrict the inclusion of more complex problems.**
>
> **A1:** We have acknowledged this limitation in Appendix A. Our decision to maintain a moderate difficulty level in DynaMath is based on two main considerations:
> 1. **Objective Focus**: DynaMath's primary goal is to evaluate the reasoning robustness of VLMs. Moderate difficulty questions highlight scenarios where VLMs can succeed with some variants but fail with others, which is crucial for improving current models. Highly complex questions might result in VLMs failing across all variants, thereby making the assessment of reasoning robustness less insightful.
> 2. **Resource Constraints**: Crafting seed questions for complex problems is resource-intensive, necessitating the expertise of human specialists, because we need to make a python program to dynamically generate new images and corresponding answers for each seed question. Medium-level questions take about 30 minutes each to design and verify, while hard-level questions require up to 1 hour, making the process more demanding.
> Considering these factors, we choose the current difficulty level, aligning with datasets like MathVerse and MathVista. In the future, we plan to develop scalable and automated pipelines using advanced foundational models to create more challenging dynamic questions for VLMs as discussed in General Response 2.
>
>
> **Q2. The selection of seed questions and their variants might introduce unintended biases. The paper doesn't deeply discuss potential limitations in the variation generation process.**
>
> **A2:** Thanks for the insightful question. We added a paragraph (Appendix A) to address the unintended biases introduced by the selection of seed questions and the potential limitations in our variation generation process. To mitigate the unintended bias, we follow previous studies [1][2] to report the dataset statistics and the source distribution of our selected seed questions.
>
> Regarding the limitation of variation generation process, we added the following discussion to our paper (Appendix A): we currently consider only individual types of variants, such as Numerical Value Variants or Function Type Variants, for each seed question. However, in many cases, it is possible to combine different types of variants, such as Color Variants and Numerical Value Variants. We will explore the integration of different variant types to further investigate the reasoning robustness of VLMs.
>
>
>
>
> **Q3. The human evaluation seems limited (only 10 participants). It is unclear if the human evaluators' backgrounds are representative, which may limit the validation of human performance.**
>
> **A3:**  To improve the validity of human evaluation, we recruited **10 additional** undergraduate and graduate students from STEM fields for human evaluators, leading to 20 students in total.  We averaged their accuracy as the human evaluation score. We found that human performance remained relatively stable and robust, with no significant differences, thereby validating that our human evaluation results are acceptable.
>
>
>
> **Q4. Generalization of the conclusion on realistic questions. The questions (especially the vision content) are created by the program, which is quite different from the figures in the application (like the exam paper; hand-written question, or photo of the book). If the model fits well on these tasks, does it mean the model performs well on the realistic scenario?**
>
> **A4:** To mitigate the gap between synthetic figures and real-world figures, we have included a Real-life Contexts Variants type in DynaMath. This variant adjusts the content of real-world scenarios, such as calendars, time-related problems, and poker-like questions, to evaluate the model’s contextual understanding and application to practical situations. However, a gap still persists due to the limited availability of open-source real-world visual math datasets, which is a common limitation across all current visual math benchmarks [1][2]. Addressing this gap represents a promising direction for future research efforts.

---

> ### Author Response · Authors · 2024-11-22
> **Author Response (part 2/2)**
>
> **Q5. Missing in-depth analysis of the performance gap between the worst-case and the standard accuracy. If we use a large amount of synthetic data in a similar manner, can the model work well also on these problems? If the performance gap can be resolved by introducing more synthetic data, the conclusion will be trivial and have no interesting news.**
>
> **A5:** The low worst-case accuracy in current VLMs can result from several factors. In the last paragraph of Section 4, we identified and analyzed five categories of errors: figure reading errors, reasoning errors, knowledge errors, calculation errors, and hallucination errors, with detailed examples provided in Appendix F. Addressing these errors is essential for enhancing the reasoning capabilities of VLMs. Our results suggest that figure reading and reasoning errors are predominant and could be prioritized for further investigation.
>
>  From a generalization theory perspective, the in-distribution performance gap can be affected by sample size, indicating that synthetic data might help mitigate some performance issues. However, our findings also emphasize the challenge of memorization phenomenon, which may not be addressed simply through an increase in synthetic data. Instead, it is important to design more effective methods to teach VLMs real reasoning skills, rather than relying on rote memorization.
>
> In summary, the primary goal of our work is to draw the community's attention to the current limitations and weaknesses of existing VLMs’ reasoning ability, along with the underlying causes of their failures. By doing so, we aim to facilitate the development and evaluation of more robust VLMs in the future.
>
>
>
> **Q6. Can we convert the existing problem into a dynamic/programmatic version?**
>
> **A6:** Yes, in DynaMath, 45.3% of the questions are derived from existing datasets like MathVista and Math-V (as indicated in Table 1). Currently, this process involves human effort. In the future, a promising direction is to use DynaMath as in-context examples or training sets to enable code LLMs/VLMs to automatically generate new seed questions and their corresponding Python programs from existing math datasets. We have discussed this in General Response 2.
>
>
>
> **Q7. Including analysis of the correlation between different types of variations and model performance will be better.**
>
> **A7:**, We added **new analysis** in **Appendix H.1 and Figure 12** to demonstrate the correlation between different variant types and reasoning robustness. We find that both GPT-4o and Qwen2-VL-72B are sensitive to certain variations including graph structure, geometric transformation, and function type. Additionally, Qwen2-VL-72B is vulnerable to symbolic substitution variants. These findings suggest directions for future improvement of these models.
>
>
> **Q8. Investigate potential relationships between training data/training techniques and performance on different variation types if possible.**
>
> **A8:** Thanks for the question. We agree that exploring how training data and methods can enhance the reasoning robustness of VLMs is an important area for future research. However, as this paper primarily focuses on benchmarking VLMs, similar to MathVista and MathVerse, and given that many SOTA open-source and closed-source models do not disclose their training data, we will leave this for future investigations.
>
> **References**
>
> [1] Zhang R, Jiang D, Zhang Y, et al. Mathverse: Does your multi-modal LLM truly see the diagrams in visual math problems?[C]//European Conference on Computer Vision. Springer, Cham, 2025: 169-186.
>
> [2] Lu P, Bansal H, Xia T, et al. MathVista: Evaluating mathematical reasoning of foundation models in visual contexts[J]. arXiv preprint arXiv:2310.02255, 2023.

---

> > ### Author Response · Authors · 2024-12-01
> >
> > Dear Reviewer,
> >
> > Thanks for your valuable comments and constructive feedback. We greatly appreciate the time and effort you have taken to review our work and provide valuable insights. In response to your suggestions, we have made significant improvements to the paper including: adding a detailed analysis of the correlation between different problem variants and reasoning robustness; conducting additional human evaluations with 10 more undergraduate and graduate students from STEM field; including a new paragraph that discusses unintended bias and potential limitations in the variation generation process of DynaMATH.
> >
> > With the discussion period closing soon, please feel free to let us know if you have any further questions or feedback. We would be sincerely grateful if you could reevaluate our work in light of our detailed responses and the revised paper.
> >
> > Sincerely,
> > Authors

---

### Author Response · Authors · 2024-11-22
**General Response 1**

We sincerely thank all the reviewers for investing their valuable time to review our work. We are encouraged to see that the dynamic nature of DynaMATH is recognized as **innovative** and **addresses a critical gap** in the existing evaluation methods (bDGf, Nqb6, Fgu5); our benchmark **complements existing ones** and **aligns well with the real-world mathematical applications**, where the problem conditions often vary (GLTU, Nqb6). Reviewers praised our **comprehensive evaluation** across diverse topics and models (bDGf, GLTU, Fgu5, Nqb6), found our **findings interesting** (Fgu5), and appreciated our **clear documentation** (bDGf).

In response to the reviewers comments, we have uploaded a revised version of our paper, where the new results and major modifications are highlighted in orange. Here are the list of the updates to the original submission:

1. Section 4: we updated the evaluation results with temperature = 0.

2. Section 4: we included few-shot CoT results on Qwen2-VL-72b (Table 2 and Table 3)

3. Section 4: we updated the human evaluation results to include a total of 20 participants.

4. Appendix A: we added more discussions on the limitations and scalability of DynaMATH.

5. Appendix D: we present more examples and variants on each problem type.

6. Appendix H: we have added more experimental results including: reasoning robustness across different variants types, problem types, and difficulty levels (Appendix H.1, H.3), in-depth failure case analysis (Appendix H.2), results with other prompt template (Appendix H.4, H.5), evaluation the robustness of DynaMATH (Appendix H.6), average accuracy variance with multiple over repeated evaluations (Appendix H.7), and more results on the circular consistency (Appendix H.8).

Besides these new results and technical discussions added above, we also made minor revisions in many aspects to further improve paper quality overall, including typo and grammar fixes, figure clarity improvements, and the addition of latest references.

---

> ### Author Response · Authors · 2024-11-22
> **General Response 2**
>
> Before delving into the specific feedback from each reviewer, we wish to first address a common point that has been raised by two reviewers: how to scale up the development of DynaMATH. We agree that it is important to scale up the design process of DynaMATH for constructing more comprehensive and challenging benchmarks. Below, we outline the primary challenges and discuss potential solutions:
>
> A key challenge in scaling DynaMATH is incorporating dynamic visual elements for each question. Unlike text-only benchmarks, our dataset includes an image for every problem with different variants (e.g., graphs, geometric shapes, function plots, real-life content). This requires careful design of the drawing program, adding significant manual effort, especially in quality control and verification, which complicates full automation.
>
> A promising solution is to leverage LLMs to automate the generation of dynamic benchmarks. LLMs have shown proficiency in generating text-based problems and writing code. It is possible to break down benchmark topics and subtopics, prompting the LLM to generate diverse problem sets and corresponding Python programs for visual elements. However, the generated problems should be dynamic, with parameterizable Python code to produce multiple image variants. To this end, DynaMATH is a valuable benchmark since our seed questions can serve as high-quality human demonstrations to guide the LLMs for this task. This LLM-assisted approach could significantly reduce manual effort. However, some human intervention will still be necessary to ensure the selection of correct and high-quality samples from LLMs.
>
> While we have to leave the LLM-assisted dynamic benchmark generation as a future work, DynaMATH can serve as a good baseline which is completely crafted by human beings, and future work on automated dynamic benchmark generation may compare to DynaMATH in terms of diversity and quality. We have included the above discussions in Appendix A in the revised paper.

---

### Author Response · Authors · 2024-11-25
**Thank you for the valuable feedback and we look forward to discussing with reviewers**

Dear Reviewers,

We would like to thank you again for your encouraging comments and constructive feedback. We are grateful to all reviewers for positively recognizing our contribution on creating a *dynamic* math benchmark for evaluating the *reasoning robustness* of VLMs. Based on your valuable comments, we have added **comprehensive additional experiments and result analysis**, including evaluation with zero temperature, CoT on open-source models, evaluations under different prompt templates, the addition of a new circular consistency metric, further human evaluations, and more in-depth analysis of results. We also clarified and addressed all questions. Your feedback has helped us to make our paper significantly stronger.

Since the discussion period is closing very soon on Nov 26, we hope you can kindly give us additional feedback and reevaluate our paper based on our responses and the revised paper. We thank you again for positively supporting our paper and we look forward to having fruitful discussions with you.

Sincerely,
DynaMath Authors

---

### Meta-Review · Area_Chair_AfRX · 2024-12-22

**Metareview:**

This work proposed a new benchmark called DynaMath, to examine the math reasoning capability and roboustness for VLMs. Drawing the inspiration from how human conduct reasoning on the math problems, the authors proposed a programmatic way of generating a set of visual reasoning question variations based on 501 seed questions spanning in different domains. These variations play an important role to study how the sutle changes to the base question would affect the final performance of VLMs. Through extensive evaluations, the authors showed that even the state-of-the-art model like GPT-4o is still not robust enough to handle the math reasoning problems. These findinds provide a new perspective about the visual reasoning capability for VLMs and shed lights to the degisn of more robust visual reasoning models.

The main contributiion of this work is the proposal of a new math reasoning benchmark and the associated extensive evaluations on a wide range of VLMs. As pointed out by the reviewers, the proposed benchmark differs from previous one in that it provides a controllable setting to make sure the image and question varients are sourced from the same type, and thus make it possible to examine the robustness of VLMs. The ACs believe this new benchmark will bring a lot benefit to the community while developing more robust visual reasoning models for math problems. There are a few weaknesses for this work, while most of them were addressed. In particular, reviewers had concerns about the generality of the proposed method to generate evaluation sample varients, and some other reviewers questioned the unintended bias for the dataset as it originates from 501 seed questions. The ACs think the response from the reviewers are mostly persuasive. However, one point to note is that the proposed benchmark is still a static one depsite the full set was generated by altering the seed one dynamically (as pointed out by Reviewer bDGf). Ideally, a true dynamic benchmarks should work like a moving target for which the VLMs cannot easily tackle by memorization. In this sense, the ACs suggest the authors to define the 'dynamic' clearly to avoid misunderstanding in the final version.

This work got four positive ratings (6,6,8,8), which clearly demonstrate its strengths and contributions to the community. As such, the ACs recommend an acceptance. Given that the proposed benchmark is the first that can support evaluating the robustness for VLMs in a controllable way, the ACs think it worth to highlight this work in the conference.

**Additional Comments On Reviewer Discussion:**

The discussions between authors and reviewers are mainly focused on the generalization of the observation on the proposed benchmark to other scenarios, the reliability of the benchmarking results for various datset settings, etc. The authors managed to provide thorough analysis and additional experimental results to demonstrate the robustness and reliability of the proposed benchmark.

---

### Decision · Program_Chairs · 2025-01-22

Accept (Poster)